# Gradient-Free Kernel Stein Discrepancy

**Matthew A. Fisher[1], Chris. J Oates[1,2]**
[1]Newcastle University, UK
[2]Alan Turing Institute, UK

## Abstract

Stein discrepancies have emerged as a powerful statistical tool, being applied to fundamental statistical problems including parameter inference, goodness-of-fit testing, and sampling. The canonical Stein discrepancies require the derivatives of a statistical model to be computed, and in return provide theoretical guarantees of convergence detection and control. However, for complex statistical models, the stable numerical computation of derivatives can require bespoke algorithmic development and render Stein discrepancies impractical. This paper focuses on posterior approximation using Stein discrepancies, and introduces a collection of non-canonical Stein discrepancies that are *gradient-free*, meaning that derivatives of the statistical model are not required. Sufficient conditions for convergence detection and control are established, and applications to sampling and variational inference are presented.

## 1  Introduction

Stein discrepancies were introduced in Gorham and Mackey [2015], as a way to measure the quality of an empirical approximation to a continuous statistical model involving an intractable normalisation constant. Rooted in *Stein's method* [Stein, 1972], the idea is to consider empirical averages of a large collection of test functions, each of which is known to integrate to zero under the statistical model. To date, test functions have been constructed by combining derivatives of the statistical model with reproducing kernels [Chwialkowski et al., 2016, Liu et al., 2016, Gorham and Mackey, 2017, Gong et al., 2021a,b], random features [Huggins and Mackey, 2018], diffusion coefficients and functions with bounded derivatives [Gorham et al., 2019], neural networks [Grathwohl et al., 2020], and polynomials [Chopin and Ducrocq, 2021]. The resulting discrepancies have been shown to be powerful statistical tools, with diverse applications including parameter inference [Barp et al., 2019, Matsubara et al., 2022], goodness-of-fit testing [Jitkrittum et al., 2017, Fernandez et al., 2020], and sampling [Liu and Lee, 2017, Chen et al., 2018, 2019, Riabiz et al., 2022, Hodgkinson et al., 2020, Fisher et al., 2021]. However, one of the main drawbacks of these existing works is the requirement that derivatives both exist and can be computed.

The use of non-differentiable statistical models is somewhat limited but includes, for example, Bayesian analyses where Laplace priors are used [Park and Casella, 2008, Ročková and George, 2018]. Much more common is the situation where derivatives exist but cannot easily be computed. In particular, for statistical models with parametric differential equations involved, one often requires different, more computationally intensive numerical methods to be used if the *sensitivities* (i.e. derivatives of the solution with respect to the parameters) are to be stably computed [Cockayne and Duncan, 2021]. For large-scale partial differential equation models, as used in finite element simulation, computation of sensitivities can increase simulation times by several orders of magnitude, if it is practical at all.

The motivation and focus of this paper is on computational methods for posterior approximation, and to this end we propose a collection of non-canonical Stein discrepancies that are *gradient free*, meaning that computation of the derivatives of the statistical model is not required. Gradient-free

37th Conference on Neural Information Processing Systems (NeurIPS 2023).

Stein operators were introduced in Han and Liu [2018] in the context of Stein variational gradient descent [Liu and Wang, 2016], but the theoretical properties of the corresponding discrepancy have yet to be investigated. General classes of Stein discrepancies were analysed in Huggins and Mackey [2018], Gorham et al. [2019], but their main results do not cover the gradient-free Stein discrepancies developed in this work, for reasons that will be explained. The combination of gradient-free Stein operators and reproducing kernels is studied in detail, to obtain discrepancies that can be explicitly computed. The usefulness of these discrepancies depends crucially on their ability to detect the convergence and non-convergence of sequences of probability measures to the posterior target, and in both directions positive results are established.

**Outline**   Gradient-free KSD (GF-KSD) is proposed and theoretically analysed in Section 2. The proposed discrepancy involves certain degrees of freedom, including a probability density denoted $q$ in the sequel, and strategies for specifying these degrees of freedom are empirically assessed in Section 3. Two applications are then explored in detail; Stein importance sampling (Section 4.1) and Stein variational inference (Section 4.2). Conclusions are drawn in Section 5.

## 2   Methods

This section contains our core methodological (Section 2.1) and theoretical (Section 2.2) development. The following notation will be used:

**Real Analytic Notation**   For a twice differentiable function $f : \mathbb{R}^d \to \mathbb{R}$, let $\partial_i f$ denote the partial derivative of $f$ with respect to its $i$th argument, let $\nabla f$ denote the gradient vector with entries $\partial_i f$, and let $\nabla^2 f$ denote the Hessian matrix with entries $\partial_i \partial_j f$. For a sufficiently regular bivariate function $f : \mathbb{R}^d \times \mathbb{R}^d \to \mathbb{R}$, let $(\partial_i \otimes \partial_j) f$ indicate the application of $\partial_i$ to the first argument of $f$, followed by the application of $\partial_j$ to the second argument. (For derivatives of other orders, the same tensor notation $\otimes$ will be used.)

**Probabilistic Notation**   Let $\mathcal{P}(\mathbb{R}^d)$ denote the set of probability distributions on $\mathbb{R}^d$. Let $\delta(x) \in \mathcal{P}(\mathbb{R}^d)$ denote an atomic distribution located at $x \in \mathbb{R}^d$. For $\pi, \pi_0 \in \mathcal{P}(\mathbb{R}^d)$, let $\pi \ll \pi_0$ indicate that $\pi$ is absolutely continuous with respect to $\pi_0$. For $\pi \in \mathcal{P}(\mathbb{R}^d)$ and $(\pi_n)_{n \in \mathbb{N}} \subset \mathcal{P}(\mathbb{R}^d)$, write $\pi_n \xrightarrow{d} \pi$ to indicate weak convergence of $\pi_n$ to $\pi$. The symbols $p$ and $q$ are reserved for probability density functions on $\mathbb{R}^d$, while $\pi$ is reserved for a generic element of $\mathcal{P}(\mathbb{R}^d)$. For convenience, the symbols $p$ and $q$ will also be used to refer to the probability distributions that these densities represent.

### 2.1   Gradient-Free Kernel Stein Discrepancy

The aim of this section is to explain how a GF-KSD can be constructed. Let $p \in \mathcal{P}(\mathbb{R}^d)$ be a target distribution of interest. Our starting point is a gradient-free Stein operator, introduced in Han and Liu [2018] in the context of Stein variational gradient descent [Liu and Wang, 2016]:

**Definition 1** (Gradient-Free Stein Operator). *For $p, q \in \mathcal{P}(\mathbb{R}^d)$ with $q \ll p$ and $\nabla \log q$ well-defined, the* gradient-free Stein operator *is defined as*

$$\mathcal{S}_{p,q} h := \frac{q}{p} \left( \nabla \cdot h + h \cdot \nabla \log q \right),$$

*acting on differentiable functions $h : \mathbb{R}^d \to \mathbb{R}^d$.*

The *Langevin* Stein operator of Gorham and Mackey [2015] is recovered when $p = q$, but when $q \neq p$ the dependence on the derivatives of $p$ is removed. The operator $\mathcal{S}_{p,q}$ can still be recognised as a *diffusion* Stein operator, being related to the infinitesimal generator of a diffusion process that leaves $p$ invariant; however, it falls outside the scope of the theoretical analysis of Huggins and Mackey [2018], Gorham et al. [2019], for reasons explained in Remark 2. It can also be viewed as a non-standard instance of the *density method* of Diaconis et al. [2004]; see Section 2 of Anastasiou et al. [2023]. The inclusion of $q$ introduces an additional degree of freedom, specific choices for which are discussed in Section 3.

**Remark 1.** *The ratio $q/p$ in Definition 1 could be viewed as an importance weight, but the construction does not fully correspond to importance sampling due to the $\nabla \log q$ term, which is $q$-dependent.*

The *Stein operator* nomenclature derives from the vanishing integral property in Proposition 1 below, which is central to Stein's method [Stein, 1972]:

**Proposition 1.** *In the setting of Definition 1, assume that $\|x\|^{d-1}q(x) \to 0$ as $\|x\| \to \infty$ and $\int \|\nabla \log q\| \, \mathrm{d}q < \infty$. Then, for any function $h : \mathbb{R}^d \to \mathbb{R}^d$ whose first derivatives exist and are bounded, it holds that $\int \mathcal{S}_{p,q} h \, \mathrm{d}p = 0$.*

All proofs are contained in Appendix A. From Proposition 1, the expectation of $\mathcal{S}_{p,q} h$ with respect to $\pi \in \mathcal{P}(\mathbb{R}^d)$ will be zero when $\pi$ and $p$ are equal; conversely, the value of such an expectation can be used to quantify the extent to which $\pi$ and $p$ are different. Consideration of multiple test functions increases the number and nature of the differences between $\pi$ and $p$ that may be detected. A *discrepancy* is obtained by specifying which test functions $h$ are considered, and then taking a supremum over the expectations associated to this set. For computational convenience we take $h$ to be contained in the unit ball of a reproducing kernel Hilbert space, as described next.

For a symmetric positive definite function $k : \mathbb{R}^d \times \mathbb{R}^d \to \mathbb{R}$, called a *kernel*, denote the associated reproducing kernel Hilbert space as $\mathcal{H}(k)$. Let $\mathcal{H}(k)^d$ denote the Cartesian product of $d$ copies of $\mathcal{H}(k)$, equipped with the inner product $\langle h, g \rangle_{\mathcal{H}(k)^d} := \sum_{i=1}^{d} \langle h_i, g_i \rangle_{\mathcal{H}(k)}$.

**Proposition 2.** *Let $\pi \in \mathcal{P}(\mathbb{R}^d)$. In the setting of Definition 1, assume there is an $\alpha > 1$ such that $\int (q/p)^\alpha \, \mathrm{d}\pi < \infty$ and $\int \|\nabla \log q\|^{\alpha/(\alpha-1)} \, \mathrm{d}\pi < \infty$. Let $k : \mathbb{R}^d \times \mathbb{R}^d \to \mathbb{R}$ be a continuously differentiable kernel such that both $k$ and its first derivatives $x \mapsto (\partial_i \otimes \partial_i)k(x,x)$, $i = 1, \dots, d$, are bounded. Then $\mathcal{S}_{p,q}$ is a bounded linear operator from $\mathcal{H}(k)^d$ to $L^1(\pi)$.*

For discrete distributions $\pi$, supported on a finite subset of $\mathbb{R}^d$, the moment conditions in Proposition 2 are automatically satisfied. For general distributions $\pi$ on $\mathbb{R}^d$, the exponent $\alpha$ can be taken arbitrarily close to 1 to enable the more stringent moment condition $\int (q/p)^\alpha \, \mathrm{d}\pi < \infty$ to hold. An immediate consequence of Proposition 2 is that Definition 2 below is well-defined.

**Definition 2** (Gradient-Free Kernel Stein Discrepancy). *For $p$, $q$, $k$ and $\pi$ satisfying the preconditions of Proposition 2, the* gradient-free KSD *is defined as*

$$\mathrm{D}_{p,q}(\pi) = \sup \left\{ \int \mathcal{S}_{p,q} h \, \mathrm{d}\pi : \|h\|_{\mathcal{H}(k)^d} \leq 1 \right\}. \tag{1}$$

The GF-KSD coincides with the canonical kernel Stein discrepancy (KSD) when $p = q$, and is thus strictly more general. Note that $\mathrm{D}_{p,q}(\pi)$ is precisely the operator norm of the linear functional $h \mapsto \int \mathcal{S}_{p,q} h \, \mathrm{d}\pi$, which exists due to Proposition 2. Most common kernels satisfy the assumptions of Proposition 2, and a particularly important example is the *inverse multi-quadric* kernel

$$k(x,y) = (\sigma^2 + \|x - y\|^2)^{-\beta}, \qquad \sigma \in (0, \infty), \ \beta \in (0, 1), \tag{2}$$

which has bounded derivatives of all orders; see Lemma 4 of Fisher et al. [2021].

The use of reproducing kernels ensures that GF-KSD can be explicitly computed; see Proposition 5 and Corollary 1 in Appendix A. A general spectral characterisation of GF-KSD is provided in Proposition 6 of the supplement, inspired by the recent work of Wynne et al. [2022]. Note that the scale of Equation (6) does not matter when one is interested in the relative performance of different $\pi \in \mathcal{P}(\mathbb{R}^d)$ as approximations of a *fixed* target $p \in \mathcal{P}(\mathbb{R}^d)$. In this sense, GF-KSD may be employed with $\tilde{p}$ in place of $p$, where $p \propto \tilde{p}/Z$ and $Z$ is a normalisation constant. This feature makes GF-KSD applicable to problems of posterior approximation, and will be exploited for both sampling and variational inference in Section 4. On the other hand, GF-KSD is not applicable to problems in which the target distribution $p_\theta$ involves a parameter $\theta$, such as estimation and composite hypothesis testing, since then the normalisation term $Z_\theta$ cannot be treated as constant.

Going beyond the discrepancies discussed in Section 1, several non-canonical discrepancies have recently been proposed based on Stein's method, for example *sliced* [Gong et al., 2021a,b], *stochastic* [Gorham et al., 2020], and *conditional* [Singhal et al., 2019] Stein discrepancies (in all cases derivatives of $p$ are required). However, only a subset of these discrepancies have been shown to enjoy important guarantees of convergence detection and control. Convergence control is critical for the posterior approximation task considered in this work, since this guarantees that minimisation of Stein discrepancy will produce a consistent approximation of the posterior target. The aim of the next section is to establish such guarantees for GF-KSD.

## 2.2 Convergence Detection and Control

The canonical KSD benefits from theoretical guarantees of convergence detection and control [Gorham and Mackey, 2017]. The aim of this section is to establish analogous guarantees for GF-KSD. To set the scene, for a Lipschitz function $f : \mathbb{R}^d \to \mathbb{R}^d$, denote its Lipschitz constant $L(f) := \sup_{x \neq y} \|f(x) - f(y)\| / \|x - y\|$. Then, for measurable $g : \mathbb{R}^d \to \mathbb{R}$, we denote the *tilted* Wasserstein distance as $\mathrm{W}_1(\pi, p; g) := \sup_{L(f) \leq 1} \left| \int fg \, \mathrm{d}\pi - \int fg \, \mathrm{d}p \right|$ whenever this expression is well-defined [Huggins and Mackey, 2018]. Note that the standard 1-Wasserstein distance $\mathrm{W}_1(\pi, p)$ is recovered when $g = 1$. There is no dominance relation between $\mathrm{W}_1(\cdot, \cdot; g)$ for different $g$; the topologies they induce are different. Tilted Wasserstein distance induces a much weaker topology than, for example, divergences such as Kullback–Leibler or Hellinger, since it does not require absolute continuity of measures.

**Theorem 1** (Convergence Detection). *Let $p, q \in \mathcal{P}(\mathbb{R}^d)$ with $q \ll p$, $\nabla \log q$ Lipschitz and $\int \|\nabla \log q\|^2 \, \mathrm{d}q < \infty$. Assume there is an $\alpha > 1$ such that the sequence $(\pi_n)_{n \in \mathbb{N}} \subset \mathcal{P}(\mathbb{R}^d)$ satisfies $\int (q/p)^\alpha \mathrm{d}\pi_n \in (0, \infty)$, $\int \|\nabla \log q\|^{\alpha/(\alpha-1)} \mathrm{d}\pi_n < \infty$, $\int \|\nabla \log q\|^{\alpha/(\alpha-1)}(q/p) \, \mathrm{d}\pi_n < \infty$, and $\int f q/p \, \mathrm{d}\pi_n < \infty$ with $f(x) = \|x\|$, for each $n \in \mathbb{N}$. Let $k$ be a kernel such that each of $k$, $(\partial_i \otimes \partial_i)k$ and $(\partial_i \partial_j \otimes \partial_i \partial_j)k$ exist, are continuous, and are bounded, for $i, j \in \{1, \ldots, d\}$. Then $\mathrm{W}_1(\pi_n, p; q/p) \to 0$ implies $\mathrm{D}_{p,q}(\pi_n) \to 0$.*

Thus the convergence of $\pi_n$ to $p$, in the sense of the tilted Wasserstein distance with $g = q/p$, is *detected* by the GF-KSD $\mathrm{D}_{p,q}$. Note that the conditions on $(\pi_n)_{n \in \mathbb{N}}$ in Theorem 1 are automatically satisfied when each $\pi_n$ has a finite support.

Despite being a natural generalisation of KSD, this GF-KSD does *not* in general provide weak convergence control in the equivalent theoretical context. Indeed, Gorham and Mackey [2017] established positive results on convergence control for distributions in $\mathcal{Q}(\mathbb{R}^d)$, the set of probability distributions on $\mathbb{R}^d$ with positive density function $q : \mathbb{R}^d \to (0, \infty)$ for which $\nabla \log q$ is Lipschitz and $q$ is *distantly dissipative*, meaning that

$$\liminf_{r \to \infty} \inf \left\{ -\frac{\langle \nabla \log q(x) - \nabla \log q(y), x - y \rangle}{\|x - y\|^2} : \|x - y\| = r \right\} > 0.$$

Under the equivalent assumptions, convergence control fails for GF-KSD in general:

**Proposition 3** (Convergence Control Fails in General). *Let $k$ be a (non-identically zero) radial kernel of the form $k(x, y) = \phi(x - y)$ for some twice continuously differentiable $\phi : \mathbb{R}^d \to \mathbb{R}$, for which the preconditions of Proposition 2 are satisfied. Then there exist $p \in \mathcal{P}(\mathbb{R}^d)$, $q \in \mathcal{Q}(\mathbb{R}^d)$ and a sequence $(\pi_n)_{n \in \mathbb{N}} \subsetneq \mathcal{P}(\mathbb{R}^d)$, also satisfying the preconditions of Proposition 2, such that $\mathrm{D}_{p,q}(\pi_n) \to 0$ and yet $\pi_n \not\xrightarrow{\mathrm{d}} p$.*

The purpose of Proposition 3 is to highlight that GF-KSD is not a trivial extension of canonical KSD; it requires a bespoke treatment. This is provided in Theorem 2, next. Indeed, to ensure that GF-KSD provides convergence control, additional condition on the tails of $q$ are required:

**Theorem 2** (Convergence Control). *Let $p \in \mathcal{P}(\mathbb{R}^d)$, $q \in \mathcal{Q}(\mathbb{R}^d)$ be such that $p$ is continuous and $\inf_{x \in \mathbb{R}^d} q(x)/p(x) > 0$. Assume there is an $\alpha > 1$ such that the sequence $(\pi_n)_{n \in \mathbb{N}} \subset \mathcal{P}(\mathbb{R}^d)$ satisfies $\int (q/p)^\alpha \mathrm{d}\pi_n < \infty$ and $\int \|\nabla \log q\|^{\alpha/(\alpha-1)}(q/p) \, \mathrm{d}\pi_n < \infty$, for each $n \in \mathbb{N}$. Let $k$ be the inverse multi-quadric kernel in Equation (2). Then $\mathrm{D}_{p,q}(\pi_n) \to 0$ implies $\pi_n \xrightarrow{\mathrm{d}} p$.*

The proof of Theorem 2 is based on carefully re-casting GF-KSD between $\pi$ and $p$ as a canonical KSD between $q$ and a transformed distribution $\bar{\pi}$ (see Proposition 7 in the supplement), then appealing to the analysis of Gorham and Mackey [2017]. Compared to the analysis of KSD in Gorham and Mackey [2017], the distant dissipativity condition now appears on $q$ (a degree of freedom), rather than on $p$ (a distribution determined by the task at hand), offering a realistic opportunity for this condition to be verified. For example, one could take a Gaussian measure $q$ that dominates the target distribution $p$ for use in GF-KSD. Nevertheless, the conditions of Theorem 2 rule out distributions $p$ that are heavy-tailed. Suitable choices for $q$ are considered in Section 3.

**Remark 2** (Related Work). *Convergence control was established for discrepancies based on general classes of Stein operator in earlier work, but the required assumptions are too stringent when applied in our context. In particular, to use Huggins and Mackey [2018] it is required that the gradient*

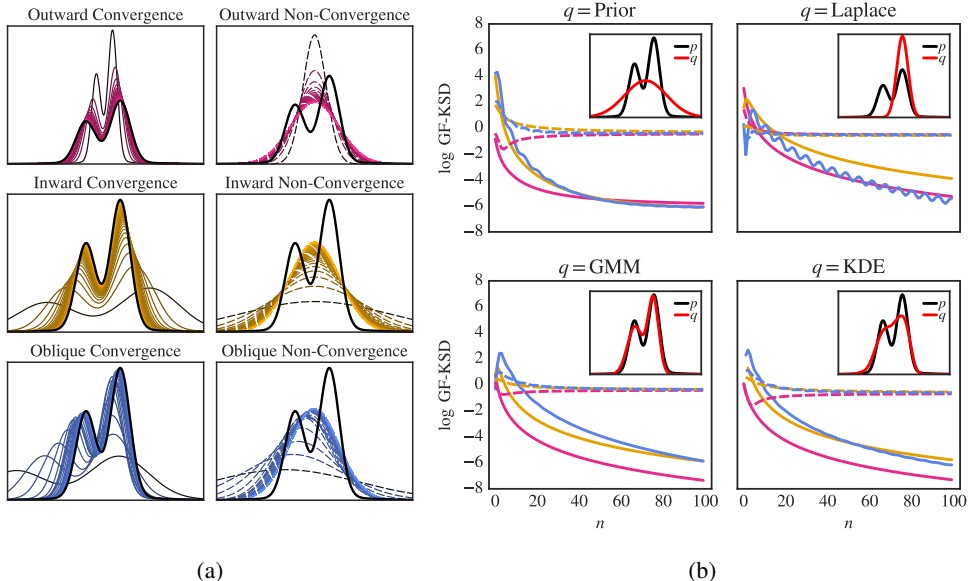

(a)                                                        (b)

Figure 1: Empirical assessment of gradient-free kernel Stein discrepancy. (a) Test sequences $(\pi_n)_{n \in \mathbb{N}}$, defined in Appendix C.1. The first column displays sequences (solid) that converge to the distributional target $p$ (black), while the second column displays sequences (dashed) which converge instead to a fixed Gaussian target. (b) Performance of gradient-free kernel Stein discrepancy, when approaches to selecting $q$ (described in the main text) are employed. The colour and style of each curve in (b) indicates which of the sequences in (a) is being considered. [Here we fixed the kernel parameters $\sigma = 1$ and $\beta = 1/2$.]

$\nabla \log(q/p)$ *is bounded*[1]*, while to use Gorham et al. [2019] it is required that $q/p$ is Lipschitz*[2]*. In our context, where $q$ must be specified in ignorance of $p$, such conditions, which require that $q$ is almost as light as $p$ in the tail, cannot be guaranteed to hold. The present paper therefore instead contributes novel analysis for the regime where $q$ may be appreciably heavier than $p$ in the tail.*

This completes our theoretical assessment, but the practical performance of GF-KSD remains to be assessed. Suitable choices for both $q$ and the kernel parameters $\sigma$ and $\beta$ are proposed and investigated in Section 3, and practical demonstrations of GF-KSD are presented in Section 4.

## 3   Implementation Detail

The purpose of this section is to empirically explore the effect of varying $q$, $\sigma$ and $\beta$, aiming to arrive at reasonable default settings. In the absence of an application-specific optimality criterion, we aim to select values that perform well (in a sense to be specified) over a range of scenarios that may be encountered. Here, to assess performance several sequences $(\pi_n)_{n \in \mathbb{N}}$ are considered, some of which converge to a specified limit $p$ and the rest of which converge to an alternative Gaussian target; see Figure 1(a). An effective discrepancy should clearly indicate which of these sequences are convergent and which are not. On this basis, recommendations for $q$ are considered in Section 3.1, and recommendations for $\sigma$ and $b$ in Section 3.2. Of course, we cannot expect default settings to perform universally well, so in Section 3.3 we highlight scenarios where our defaults may fail. Python code to reproduce the experiments reported below can be downloaded at [blinded].

### 3.1   Choice of $q$

In what follows we cast $q$ as an approximation of $p$, aiming to inherit the desirable performance of canonical KSD for which $q$ and $p$ are equal. The task to which the discrepancy is being applied will,

---

[1]To see this, take $A = q/p$ in Theorem 3.2 of Huggins and Mackey [2018].
[2]To see this, take $m = (q/p)I$ in Theorem 7 and Proposition 8 of Gorham et al. [2019].

in practice, constrain the nature and form of the distributions $q$ that can be implemented. For expository purposes (only), the following qualitatively distinct approaches to choosing $q$ are considered:

- `Prior`  In Bayesian settings where $p$ is a posterior distribution, selecting $q$ to be the prior distribution ensures that the condition $\inf_{x \in \mathbb{R}^d} q(x)/p(x) > 0$ is satisfied.

- `Laplace`  If the target $p$ can be differentiated, albeit at a possibly high computational cost, it may be practical to construct a Laplace approximation $q$ to the target [Gelman et al., 2013].

- `GMM`  One could take $q$ to be a Gaussian mixture model fitted to approximate samples from $p$, representing a more flexible alternative to `Laplace`.

- `KDE`  Additional flexibility can be obtained by employing a kernel density estimator as a non-parametric alternative to `GMM`.

Of course, there is a circularity to `GMM` and `KDE` which renders these methods impractical in general. The specific details of how each of the $q$ were constructed are contained in the code that accompanies this paper, but the resulting $q$ are displayed as insets in Figure 1(b). The performance of GF-KSD with these different choices of $q$ is also displayed in Figure 1(b). It was observed that all four choices of $q$ produced a discrepancy that could detect convergence of $\pi_n$ to $p$, though the detection of convergence was less clear for `Prior` due to slower convergence of the discrepancy to 0 as $n$ was increased. On the other hand, all approaches were able to clearly detect non-convergence to the target. That `Laplace` performed comparably with `GMM` and `KDE` was surprising, given that the target $p$ is not well-approximated by a single Gaussian component. These results are for a specific choice of target $p$, but in Appendix C.5.1 a range of $p$ are considered and similar conclusions are obtained. Section 3.3 explores how challenging $p$ must be before the convergence detection and control properties associated to `Laplace` fail.

### 3.2  Choice of $\sigma$ and $\beta$

For the investigation in Section 3.1 the parameters of the inverse multi-quadric kernel (2) were fixed to $\sigma = 1$ and $\beta = 1/2$, the latter being the midpoint of the permitted range $\beta \in (0, 1)$. In general, care in the selection of these parameters may be required. The parameter $\sigma$ captures the scale of the data, and thus standardisation of the data may be employed to arrive at $\sigma = 1$ as a natural default. In this paper (with the exception of Section 4.2) the standardisation $x \mapsto C^{-1} x$ was performed, where $C$ is the covariance matrix of the approximating distribution $q$ being used. In Appendix C.5.2 we reproduce the investigation of Section 3.1 using a range of values for $\sigma$ and $\beta$; these results indicate the performance of GF-KSD is remarkably insensitive to perturbations around $(\sigma, \beta) = (1, 1/2)$.

### 3.3  Avoidance of Failure Modes

GF-KSD is not a silver bullet, and there are a number of specific failure modes that care may be required to avoid. The four main failure modes are illustrated in Figure 2. These are as follows: (a) $q$ is substantially heavier than $p$ in a tail; (b) $q$ is substantially lighter than $p$ in a tail; (c) the dimension $d$ is too high; (d) $p$ has well-separated high-probability regions. Under both (a) and (c), convergence detection can fail, either because theoretical conditions are violated or because the terms $\pi_n$ must be extremely close to $p$ before convergence begins to be detected. Under (b), the values of GF-KSD at small $n$ can mislead. Point (d) is a well-known pathology of all score-based methods; see Wenliang and Kanagawa [2021], Liu et al. [2023]. These four failure modes inform our recommended usage of GF-KSD, summarised next.

**Summary of Recommendations**  Based on the investigation just reported, the default settings we recommend are `Laplace` with $\sigma = 1$ (post-standardisation) and $\beta = 1/2$. Although not universally applicable, `Laplace` does not require samples from $p$, and has no settings that must be user-specified. Thus we recommend the use of `Laplace` in situations where a Laplace approximation can be justified and computed. If `Laplace` not applicable, then one may attempt to construct an approximation $q$ using techniques available for the task at hand (e.g. in a Bayesian setting, one may obtain $q$ via variational inference, or via inference based on an approximate likelihood). These recommended settings will be used for the application presented in Section 4.1, next.

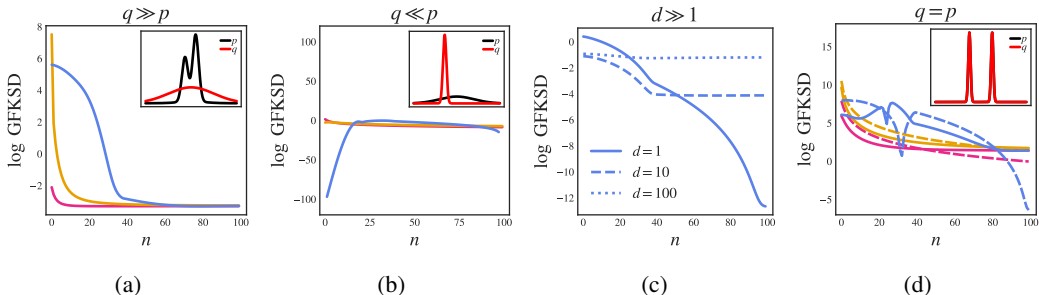

Figure 2: Failure modes: (a) $q$ is substantially heavier than $p$ in a tail; (b) $q$ is substantially lighter than $p$ in a tail; (c) the dimension $d$ is too high; (d) $p$ has separated high-probability regions. [For each of (a), (b) and (d), the colour and style of the curves refers to the same sense of convergence or non-convergence (outward/inward/oblique) presented in Figure 1, and we plot the logarithm of the gradient-free kernel Stein discrepancy as a function of the index $n$ of the sequence $(\pi_n)_{n \in \mathbb{N}}$. For (c) we consider convergent sequences $(\pi_n)_{n \in \mathbb{N}}$ of distributions on $\mathbb{R}^d$.]

## 4 Applications

To demonstrate potential uses of GF-KSD, two applications to posterior approximation are now presented; Stein importance sampling (Section 4.1) and Stein variational inference using measure transport (Section 4.2). In each case, we extend the applicability of existing algorithms to statistical models for which certain derivatives of $p$ are either expensive or non-existent.

### 4.1 Gradient-Free Stein Importance Sampling

Stein importance sampling [Liu and Lee, 2017, Hodgkinson et al., 2020] operates by first sampling independently from a tractable approximation of the target $p$ and then correcting the bias in the samples so-obtained. To date, applications of Stein importance sampling have been limited to instances where the statistical model $p$ can be differentiated; our contribution is to remove this requirement. In what follows we analyse Stein importance sampling in which independent samples $(x_n)_{n \in \mathbb{N}}$ are generated from the same approximating distribution $q$ that is employed within GF-KSD:

**Theorem 3** (Gradient-Free Stein Importance Sampling). *Let $p \in \mathcal{P}(\mathbb{R}^d)$, $q \in \mathcal{Q}(\mathbb{R}^d)$ be such that $p$ is continuous and $\inf_{x \in \mathbb{R}^d} q(x)/p(x) > 0$. Suppose that $\int \exp\{\gamma \|\nabla \log q\|^2\} \, \mathrm{d}q < \infty$ for some $\gamma > 0$. Let $k$ be the inverse multi-quadric kernel in Equation (2). Let $(x_n)_{n \in \mathbb{N}}$ be independent samples from $q$. To the sample, assign optimal weights*

$$w^* \in \arg\min \left\{ \mathrm{D}_{p,q} \left( \sum_{i=1}^n w_i \delta(x_i) \right) : 0 \leq w_1, \ldots, w_n, \; w_1 + \cdots + w_n = 1 \right\}.$$

*Then $\pi_n := \sum_{i=1}^n w_i^* \delta(x_i)$ satisfies $\pi_n \xrightarrow{\mathrm{d}} p$ almost surely as $n \to \infty$.*

The proof builds on earlier work in Riabiz et al. [2022]. Note that the optimal weights $w^*$ can be computed without the normalisation constant of $p$, by solving a constrained quadratic programme at cost $O(n^3)$.

As an illustration, we implemented gradient-free Stein importance sampling to approximate a posterior arising from a discretely observed Lotka–Volterra model

$$\dot{u}(t) = \alpha u(t) - \beta u(t)v(t), \qquad \dot{v}(t) = -\gamma v(t) + \delta u(t)v(t), \qquad (u(0), v(0)) = (u_0, v_0),$$

with independent log-normal observations with covariance matrix $\mathrm{diag}(\sigma_1^2, \sigma_2^2)$. The parameters to be inferred are $\{\alpha, \beta, \gamma, \delta, u_0, v_0, \sigma_1, \sigma_2\}$ and therefore $d = 8$. The data analysed are due to Hewitt [1921], and full details are contained in Appendix C.3. The direct application of Stein importance sampling to this task requires the numerical calculation of *sensitivities* of the differential equation at each of the $n$ samples that are to be re-weighted. Aside from simple cases where automatic differentiation or adjoint methods can be used, the stable computation of sensitivities can form a major computational bottleneck; see [Riabiz et al., 2022]. In contrast, our approach required a

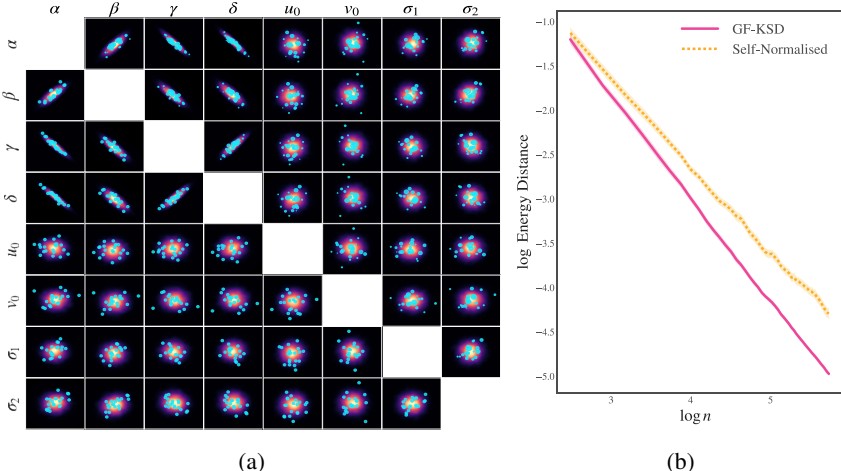

(a)                                                                    (b)

Figure 3: Gradient-Free Stein Importance Sampling: (a) The lower triangular panels display $n = 20$ independent (biased) samples from the Laplace approximation $q$, while the upper triangular panels display the same number of re-weighted samples obtained using gradient-free Stein importance sampling. [Samples are shown in blue, with their size proportional to the square of their weight, to aid visualisation. The shaded background indicates the high probability regions of $p$, the target.] (b) The approximation quality, as a function of the number $n$ of samples from $q$, is measured as the energy distance between the approximation and the target. [The solid line corresponds to the output of gradient-free Stein importance sampling, while the dashed line corresponds to the output of self-normalised importance sampling. Standard error regions are shaded.]

fixed number of gradient computations to construct a Laplace approximation[3], independent of the number $n$ of samples required; see Appendix C.3 for detail. In the lower triangular portion of Figure 3(a), biased samples from the Laplace approximation $q$ ($\neq p$) are displayed, while in the upper triangular portion the same samples are re-weighted using gradient-free Stein importance sampling to form a consistent approximation of $p$. A visual reduction in bias and improvement in approximation quality can be observed. As a baseline against which to assess the quality of our approximation, we consider self-normalised importance sampling; i.e. the approximation with weights $\bar{w}$ such that $\bar{w}_i \propto p(x_i)/q(x_i)$. Figure 3(b) reports the accuracy of the approximations to $p$ as quantified using *energy distance* [Cramér, 1928]. These results indicate that the approximations produced using gradient-free Stein importance sampling improve on those constructed using self-normalised importance sampling. This may be explained by the fact that the optimal weights $w^*$ attempt to mitigate both bias due to $q \neq p$ and Monte Carlo error, while the weights $\bar{w}$ only address the bias due to $q \neq p$, and do not attempt to mitigate error due to the randomness in Monte Carlo. Additional experiments in Appendix C.5.3 confirm that gradient-free Stein importance sampling achieves comparable performance with gradient-based Stein importance sampling in regimes where the Laplace approximation can be justified.

Although our recommended default settings for GF-KSD were successful in this example, an interesting theoretical question would be to characterise an optimal choice of $q$ in this context. This appears to be a challenging problem but we hope to address it in future work. In addition, although we focused on Stein importance sampling, our methodology offers the possibility to construct gradient-free versions of other related algorithms, including the *Stein points* algorithms of Chen et al. [2018, 2019], and the *Stein thinning* algorithm of Riabiz et al. [2022].

## 4.2 Stein Variational Inference Without Second-Order Gradient

Stein discrepancy was proposed as a variational objective in Ranganath et al. [2016] and has demonstrated comparable performance with the traditional Kullback–Leibler objective in certain application areas, whilst abolishing the requirement that the variational family is absolutely continuous with

---

[3]In this case, 49 first order derivatives were computed, of which 48 were on the optimisation path and 1 was used to construct a finite difference approximation to the Hessian.

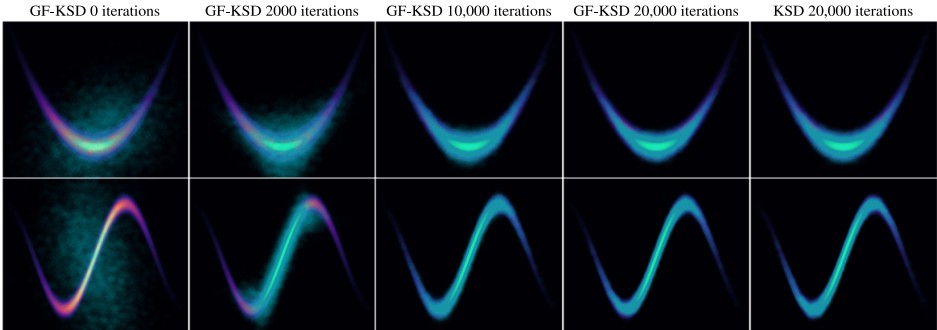

| GF-KSD 0 iterations | GF-KSD 2000 iterations | GF-KSD 10,000 iterations | GF-KSD 20,000 iterations | KSD 20,000 iterations |

Figure 4: Stein Variational Inference Without Second Order Gradient: The top row concerns approximation of a distributional target $p$ that is "banana" shaped, while the bottom row concerns a "sinusoidal" target. The first four columns depict the variational approximation $\pi_{\theta_m}$ to $p$ constructed using gradient descent applied to gradient-free kernel Stein discrepancy (i.e. first order derivatives of $p$ required) along the stochastic optimisation sample path ($m \in \{0, 2 \times 10^3, 10^4, 2 \times 10^4\}$), while the final column reports the corresponding approximation ($m = 2 \times 10^4$) constructed using standard kernel Stein discrepancy (i.e. second order derivatives of $p$ required).

respect to the statistical model [Fisher et al., 2021]. This offers an exciting, as yet largely unexplored opportunity to construct flexible variational families outside the conventional setting of normalising flows (which are constrained to be diffeomorphisms of $\mathbb{R}^d$). However, gradient-based stochastic optimisation of the canonical Stein discrepancy objective function means that second-order derivatives of the statistical model are required. GF-KSD reduces the order of derivatives that are required from second-order to first-order, and in many cases (such as when differential equations appear in the statistical model) this will correspond to a considerable reduction in computational cost.

In what follows we fix a reference distribution $R \in \mathcal{P}(\mathbb{R}^d)$ and a parametric class of maps $T^\theta : \mathbb{R}^d \to \mathbb{R}^d$, $\theta \in \mathbb{R}^p$, and consider the variational family $(\pi_\theta)_{\theta \in \mathbb{R}^p} \subset \mathcal{P}(\mathbb{R}^d)$ whose elements $\pi_\theta := T^\theta_\# R$ are the *pushforwards* of $R$ through the maps $T^\theta$, $\theta \in \mathbb{R}^p$. The aim is to use stochastic optimisation to minimise $\theta \mapsto D_{p,q}(\pi_\theta)$ (or, equivalently, any strictly increasing transformation thereof). For this purpose a low-cost unbiased estimate of the gradient of the objective function is required; the details and sufficient theoretical conditions are contained in Appendix B.

As an interesting methodological extension, that departs from the usual setting of stochastic optimisation, here we consider interlacing stochastic optimisation over $\theta$ and the selection of $q$, leveraging the current value $\theta_m$ on the optimisation path to provide a natural candidate $\pi_{\theta_m}$ for $q$ in this context[4]. Thus, for example, a vanilla stochastic gradient descent routine becomes $\theta_{m+1} = \theta_m - \epsilon \left. \nabla_\theta D_{p,\pi_{\theta_m}}(\pi_\theta)^2 \right|_{\theta=\theta_m}$ for some learning rate $\epsilon > 0$. (In this iterative setting, to ensure the variational objective remains fixed, we do not perform the standardisation of the data described in Section 3.2.)

To assess the performance of GF-KSD in this context, we re-instantiated an experiment from Fisher et al. [2021]. The results, in Figure 4, concern the approximation of "banana" and "sinusoidal" distributions in dimension $d = 2$, and were obtained using the reference distribution $R = \mathcal{N}(0, 2I)$ and taking $T^\theta$ to be the *inverse autoregressive flow* of Kingma et al. [2016]. These are both toy problems, which do not themselves motivate our methodological development, but do enable us to have an explicit ground truth to benchmark performance against. Full experimental detail is contained in Appendix C.4; we highlight that *gradient clipping* was used, both to avoid extreme values of $q/p$ encountered on the optimisation path, and to accelerate the optimisation itself [Zhang et al., 2019]. The rightmost panel depicts the result of performing variational inference with the standard KSD objective functional. It is interesting to observe that GF-KSD leads to a similar performance in both examples, with the caveat that stochastic optimisation was more prone to occasional failure when GF-KSD was used. The development of a robust optimisation technique in this context requires care and a detailed empirical assessment, and is left as a promising avenue for further research.

---

[4]If the density of $\pi_{\theta_m}$ is not available, for example because $T$ is a complicated mapping, it can be consistently estimated using independent samples from $T^{\theta_m}_\#$.

# 5  Conclusion

In this paper GF-KSD was proposed and studied. Theoretical and empirical results support the use of GF-KSD in settings where an initial approximation to the distributional target can readily be constructed, and where the distributional target itself does not contain distant high probability regions, but poor performance can occur outside this context. Nevertheless, for many statistical analyses the principal challenge is the cost of evaluating the statistical model and its derivatives, rather than the complexity of the target itself, and in these settings the proposed discrepancy has the potential to be usefully employed. The focus of this work was on posterior approximation, with illustrative applications to sampling and variational inference being presented. A natural extension to this work would involve a systematic empirical assessment of the performance of GF-KSD across a broad range of applied contexts. However, we note that GF-KSD is not applicable to problems such as estimation and composite hypothesis testing, where the target $p_\theta$ ranges over a parametric model class, and alternative strategies will be required to circumvent gradient computation in that context.

**Acknowledgements**  MAF was supported by EP/W522387/1.  CJO was supported by EP/W019590/1.  The authors are grateful to François-Xavier Briol, Jon Cockayne, Jeremias Knoblauch, Lester Mackey, Marina Riabiz, Rob Salomone, Leah South, and George Wynne for insightful comments on an early draft of the manuscript, as well as to the Reviewers at NeurIPS.

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

# Appendices

These appendices supplement the paper *Gradient Free Kernel Stein Discrepancy*. The proofs for theoretical results stated in the main text are contained in Appendix A. Theoretical analysis of stochastic gradient estimators for use in the context of Stein Variational Inference is contained in Appendix B, as advertised in Section 4.2 of the main text. All additional details related to empirical assessment are contained in Appendix C.

## A   Proof of Results in the Main Text

This appendix contains proofs for all novel theoretical results reported in the main text.

*Proof of Proposition 1.* First we show that the integral in the statement is well-defined. Since $h$ and its first derivatives are bounded, we can define the constants $C_0 := \sup\{\|h(x)\| : x \in \mathbb{R}^d\}$ and $C_1 := \sup\{|\nabla \cdot h(x)| : x \in \mathbb{R}^d\}$. Then

$$\int |\mathcal{S}_{p,q} h| \, \mathrm{d}p = \int \left| \frac{q}{p} [\nabla \cdot h + h \cdot \nabla \log q] \right| \, \mathrm{d}p = \int |\nabla \cdot h + h \cdot \nabla \log q| \, \mathrm{d}q$$

$$\leq C_1 + C_0 \int \|\nabla \log q\| \, \mathrm{d}q < \infty,$$

so indeed the integral is well-defined. Now, let $B_r = \{x \in \mathbb{R}^d : \|x\| \leq r\}$, $S_r = \{x \in \mathbb{R}^d : \|x\| = r\}$, and $t(r) := \sup\{q(x) : x \in S_r\}$, so that by assumption $r^{d-1} t(r) \to 0$ as $r \to \infty$. Let $\mathbb{1}_B(x) = 1$ if $x \in B$ and $0$ if $x \notin B$. Then $\int \mathcal{S}_{p,q} h \, \mathrm{d}p = \lim_{r \to \infty} \int \mathbb{1}_{B_r} \mathcal{S}_{p,q} h \, \mathrm{d}p$ and, from the divergence theorem,

$$\int \mathbb{1}_{B_r} \mathcal{S}_{p,q} h \, \mathrm{d}p = \int \mathbb{1}_{B_r} \frac{q}{p} [\nabla \cdot h + h \cdot \nabla \log q] \, \mathrm{d}p$$

$$= \int_{B_r} [q \nabla \cdot h + h \cdot \nabla q] \, \mathrm{d}x$$

$$= \int_{B_r} \nabla \cdot (qh) \, \mathrm{d}x$$

$$= \oint_{S_r} qh \cdot n \, \mathrm{d}x \leq t(r) \times \frac{2\pi^{d/2}}{\Gamma(d/2)} r^{d-1} \overset{r \to \infty}{\to} 0,$$

as required. $\qquad\square$

*Proof of Proposition 2.* Since $k$ and its first derivatives are bounded, we can set

$$C_0^k := \sup_{x \in \mathbb{R}^d} \sqrt{k(x,x)}, \qquad C_1^k := \sup_{x \in \mathbb{R}^d} \sqrt{\sum_{i=1}^d (\partial_i \otimes \partial_i) k(x,x)}.$$

From Cauchy–Schwarz and the reproducing property, for any $f \in \mathcal{H}(k)$ it holds that $|f(x)| = |\langle f, k(\cdot, x) \rangle_{\mathcal{H}(k)}| \leq \|f\|_{\mathcal{H}(k)} \|k(\cdot, x)\|_{\mathcal{H}(k)} = \|f\|_{\mathcal{H}(k)} \sqrt{\langle k(\cdot, x), k(\cdot, x) \rangle_{\mathcal{H}(k)}} = \|f\|_{\mathcal{H}(k)} \sqrt{k(x,x)}$. Furthermore, using the fact that $k$ is continuously differentiable, it can be shown that $|\partial_i f(x)| \leq \|f\|_{\mathcal{H}(k)} \sqrt{(\partial_i \otimes \partial_i) k(x,x)}$; see Corollary 4.36 of Steinwart and Christmann [2008]. As a consequence, for all $h \in \mathcal{H}(k)^d$ we have that, for all $x \in \mathbb{R}^d$,

$$\|h(x)\| = \sqrt{\sum_{i=1}^d h_i(x)^2} \leq \sqrt{\sum_{i=1}^d k(x,x) \|h_i\|_{\mathcal{H}(k)}^2} = C_0^k \|h\|_{\mathcal{H}(k)^d}$$

$$|\nabla \cdot h(x)| = \left| \sum_{i=1}^d \partial_{x_i} h_i(x) \right| \leq \left| \sum_{i=1}^d \sqrt{(\partial_i \otimes \partial_i) k(x,x)} \|h_i\|_{\mathcal{H}(k)} \right| \leq C_1^k \|h\|_{\mathcal{H}(k)^d}.$$

To use analogous notation as in the proof of Proposition 1, set $C_0 := C_0^k \|h\|_{\mathcal{H}(k)^d}$ and $C_1 := C_1^k \|h\|_{\mathcal{H}(k)^d}$. Then, using Hölder's inequality and the fact that $(a+b)^\beta \leq 2^{\beta-1}(a^\beta + b^\beta)$ with $\beta = \alpha/(\alpha-1)$, we have

$$\|\mathcal{S}_{p,q} h\|_{L^1(\pi)} = \int |\mathcal{S}_{p,q} h| \, \mathrm{d}\pi = \int \left| \frac{q}{p} [\nabla \cdot h + h \cdot \nabla \log q] \right| \, \mathrm{d}\pi$$

$$\leq \left( \int \left( \frac{q}{p} \right)^\alpha \mathrm{d}\pi \right)^{\frac{1}{\alpha}} \left( \int (\nabla \cdot h + h \cdot \nabla \log q)^{\frac{\alpha}{\alpha-1}} \, \mathrm{d}\pi \right)^{\frac{\alpha-1}{\alpha}}$$

$$\leq 2^{\frac{1}{\alpha}} \left( \int \left( \frac{q}{p} \right)^\alpha \mathrm{d}\pi \right)^{\frac{1}{\alpha}} \left( C_1^{\frac{\alpha}{\alpha-1}} + C_0^{\frac{\alpha}{\alpha-1}} \int \|\nabla \log q\|^{\frac{\alpha}{\alpha-1}} \, \mathrm{d}\pi \right)^{\frac{\alpha-1}{\alpha}}$$

$$\leq 2^{\frac{1}{\alpha}} \|h\|_{\mathcal{H}(k)^d} \left( \int \left( \frac{q}{p} \right)^\alpha \mathrm{d}\pi \right)^{\frac{\alpha-1}{\alpha}} \left( (C_1^k)^{\frac{\alpha}{\alpha-1}} + (C_0^k)^{\frac{\alpha}{\alpha-1}} \int \|\nabla \log q\|^{\frac{\alpha}{\alpha-1}} \, \mathrm{d}\pi \right)^{\frac{\alpha-1}{\alpha}}$$

as required. $\qquad\square$

To prove obtain explicit computable formulae for the GF-KSD, two intermediate results are required:

**Proposition 4.** *Let $k$ and $\pi$ satisfy the preconditions of Proposition 2. Then the function*

$$\mathbb{R}^d \ni x \mapsto f(x) := \frac{q(x)}{p(x)} [\nabla_x k(x, \cdot) + k(x, \cdot) \nabla \log q(x)] \tag{3}$$

*takes values in $\mathcal{H}(k)^d$, is Bochner $\pi$-integrable and, thus, $\xi := \int f \, \mathrm{d}\pi \in \mathcal{H}(k)^d$.*

*Proof.* Since $k$ has continuous first derivatives $x \mapsto (\partial_i \otimes \partial_i)k(x,x)$, Lemma 4.34 of Steinwart and Christmann [2008] gives that $(\partial_i \otimes 1)k(x, \cdot) \in \mathcal{H}(k)$, and, thus $f \in \mathcal{H}(k)^d$. Furthermore, $f : \mathbb{R}^d \to \mathcal{H}(k)^d$ is Bochner $\pi$-integrable since

$$\int \|f(x)\|_{\mathcal{H}(k)^d} \, \mathrm{d}\pi(x) = \int \frac{q(x)}{p(x)} \|\nabla_x k(x, \cdot) + k(x, \cdot) \nabla \log q(x)\|_{\mathcal{H}(k)^d} \mathrm{d}\pi(x)$$

$$\leq \left( \int \left( \frac{q}{p} \right)^\alpha \mathrm{d}\pi \right)^{\frac{1}{\alpha}} \left( \int \|\nabla_x k(x, \cdot) + k(x, \cdot) \nabla \log q(x)\|_{\mathcal{H}(k)^d}^{\frac{\alpha}{\alpha-1}} \, \mathrm{d}\pi(x) \right)^{\frac{\alpha-1}{\alpha}}$$

$$\leq 2^{\frac{1}{\alpha}} \left( \int \left( \frac{q}{p} \right)^\alpha \mathrm{d}\pi \right)^{\frac{1}{\alpha}} \left( (C_1^k)^{\frac{\alpha}{\alpha-1}} + (C_0^k)^{\frac{\alpha}{\alpha-1}} \int \|\nabla \log q\|^{\frac{\alpha}{\alpha-1}} \, \mathrm{d}\pi \right)^{\frac{\alpha-1}{\alpha}} < \infty$$

where we have employed the same $C_0^k$ and $C_1^k$ notation as used in the proof of Proposition 2. Thus, from the definition of the Bochner integral, $\xi = \int f \, \mathrm{d}\pi$ exists and is an element of $\mathcal{H}(k)^d$. $\qquad\square$

**Proposition 5.** *Let $k$ and $\pi$ satisfy the preconditions of Proposition 2. Then*

$$\mathrm{D}_{p,q}(\pi)^2 = \iint \frac{q(x)}{p(x)} \frac{q(y)}{p(y)} k_q(x,y) \, \mathrm{d}\pi(x) \mathrm{d}\pi(y) \tag{4}$$

*where*

$$k_q(x,y) = \nabla_x \cdot \nabla_y k(x,y) + \langle \nabla_x k(x,y), \nabla_y \log q(y) \rangle + \langle \nabla_y k(x,y), \nabla_x \log q(x) \rangle$$
$$+ k(x,y) \langle \nabla_x \log q(x), \nabla_y \log q(y) \rangle. \tag{5}$$

*Proof.* Let $f$ be as in Equation (3). From Proposition 4, $\xi = \int f \, \mathrm{d}\pi \in \mathcal{H}(k)^d$. Moreover, since $f$ is Bochner $\pi$-integrable and $Tf = \langle h, f \rangle_{\mathcal{H}(k)^d}$ is a continuous linear functional on $\mathcal{H}(k)^d$, from basic properties of Bochner integrals we have $T\xi = T \int f \, \mathrm{d}\pi = \int Tf \, \mathrm{d}\pi$. In particular,

$$\langle h, \xi \rangle_{\mathcal{H}(k)^d} = \left\langle h, \int \frac{q(x)}{p(x)} [\nabla_x k(x, \cdot) + k(x, \cdot) \nabla \log q(x)] \, \mathrm{d}\pi(x) \right\rangle_{\mathcal{H}(k)^d}$$

$$= \int \frac{q(x)}{p(x)} \left[ \nabla_x \langle h, k(x, \cdot) \rangle_{\mathcal{H}(k)^d} + \langle h, k(x, \cdot) \rangle_{\mathcal{H}(k)^d} \nabla \log q(x) \right] \, \mathrm{d}\pi(x)$$

$$= \int \frac{q(x)}{p(x)} \left[ \nabla \cdot h(x) + h(x) \cdot \nabla \log q(x) \right] \, \mathrm{d}\pi(x) = \int \mathcal{S}_{p,q} h \, \mathrm{d}\pi(x)$$

which shows that $\xi$ is the Riesz representer of the bounded linear functional $h \mapsto \int \mathcal{S}_{p,q} h \, \mathrm{d}\pi$ on $\mathcal{H}(k)^d$. It follows from Cauchy–Schwarz that the (squared) operator norm of this functional is

$$\mathrm{D}_{p,q}(\pi)^2 = \|\xi\|^2_{\mathcal{H}(k)^d} = \langle \xi, \xi \rangle_{\mathcal{H}(k)^d} = \left\langle \int \frac{q(x)}{p(x)} \left[ \nabla_x k(x, \cdot) + k(x, \cdot) \nabla \log q(x) \right] \, \mathrm{d}\pi(x), \right.$$
$$\left. \int \frac{q(y)}{p(y)} \left[ \nabla_y k(y, \cdot) + k(y, \cdot) \nabla \log q(y) \right] \, \mathrm{d}\pi(y) \right\rangle_{\mathcal{H}(k)^d}$$
$$= \iint \frac{q(x)}{p(x)} \frac{q(y)}{p(y)} k_q(x, y) \, \mathrm{d}\pi(x) \mathrm{d}\pi(y)$$

as claimed. $\qquad\square$

For concreteness, we instantiate Proposition 5 in the specific case of the inverse multi-quadric kernel, since this is the kernel that we recommend in Section 2.2:

**Corollary 1** (Explicit Form). *For $p$, $q$, and $\pi$ satisfying the preconditions of Proposition 2, and $k$ the inverse multi-quadric kernel in Equation* (2)*, we have that*

$$\mathrm{D}_{p,q}(\pi)^2 = \iint \frac{q(x)q(y)}{p(x)p(y)} \left\{ \frac{4\beta(\beta+1)\|x-y\|^2}{(\sigma^2 + \|x-y\|^2)^{\beta+2}} + 2\beta \left[ \frac{d + \langle \nabla \log q(x) - \nabla \log q(y), x - y \rangle}{(\sigma^2 + \|x-y\|^2)^{1+\beta}} \right] \right.$$
$$\left. + \frac{\langle \nabla \log q(x), \nabla \log q(y) \rangle}{(\sigma^2 + \|x-y\|^2)^\beta} \right\} \, \mathrm{d}\pi(x)\mathrm{d}\pi(y) \qquad (6)$$

*Proof of Corollary 1.* First we compute derivatives of the kernel $k$ in Equation (2):

$$\nabla_x k(x, y) = -\frac{2\beta}{(\sigma^2 + \|x-y\|^2)^{\beta+1}} (x - y)$$

$$\nabla_y k(x, y) = \frac{2\beta}{(\sigma^2 + \|x-y\|^2)^{\beta+1}} (x - y)$$

$$\nabla_x \cdot \nabla_y k(x, y) = -\frac{4\beta(\beta+1)\|x-y\|^2}{(\sigma^2 + \|x-y\|^2)^{\beta+2}} + \frac{2\beta d}{(\sigma^2 + \|x-y\|^2)^{\beta+1}}$$

Letting $u(x) := \nabla \log q(x)$ form a convenient shorthand, we have that

$$k_q(x, y) := \nabla_x \cdot \nabla_y k(x, y) + \langle \nabla_x k(x, y), u(y) \rangle + \langle \nabla_y k(x, y), u(x) \rangle + k(x, y) \langle u(x), u(y) \rangle$$
$$= -\frac{4\beta(\beta+1)\|x-y\|^2}{(\sigma^2 + \|x-y\|^2)^{\beta+2}} + 2\beta \left[ \frac{d + \langle u(x) - u(y), x - y \rangle}{(\sigma^2 + \|x-y\|^2)^{1+\beta}} \right] + \frac{\langle u(x), u(y) \rangle}{(\sigma^2 + \|x-y\|^2)^\beta}$$

which, combined with Proposition 5, gives the result. $\qquad\square$

In addition to the results in the main text, here we present a spectral characterisation of GF-KSD. The following result was inspired by an impressive recent contribution to the literature on kernel Stein discrepancy due to Wynne et al. [2022], and our (informal) proof is based on an essentially identical argument:

**Proposition 6** (Spectral Characterisation). *Consider a positive definite isotropic kernel $k$, and recall that Bochner's theorem guarantees $k(x, y) = \int e^{-i\langle s, x-y \rangle} \mathrm{d}\mu(s)$ for some $\mu \in \mathcal{P}(\mathbb{R}^d)$. Then, under regularity conditions that we leave implicit,*

$$\mathrm{D}_{p,q}(\pi)^2 = \int \left\| \int \frac{1}{p(x)} \left\{ e^{-i\langle s, x \rangle} \nabla q(x) - ise^{-i\langle s, x \rangle} q(x) \right\} \mathrm{d}\pi(x) \right\|_{\mathbb{C}}^2 \mathrm{d}\mu(s). \qquad (7)$$

The Fourier transform $\widehat{\nabla q}$ of $\nabla q$ is defined as $\int e^{-i\langle s, x \rangle} \nabla q(x) \, \mathrm{d}x$, and a basic property of the Fourier transform is that the transform of a derivative can be computed using the expression $is \int e^{-i\langle s, x \rangle} q(x) \, \mathrm{d}x$. This implies that the inner integral in (7) vanishes when $\pi$ and $p$ are equal. Thus we can interpret GF-KSD as a quantification of the uniformity of $\mathrm{d}\pi/\mathrm{d}p$, with a weighting function based on the Fourier derivative identity with regard to $\nabla q$.

*Proof of Proposition 6.* From direct calculation, and assuming derivatives and integrals can be interchanged, we have that

$$\nabla_x k(x,y) = -\int i s e^{-i\langle s, x-y\rangle}\, \mathrm{d}\mu(s), \tag{8}$$

$$\nabla_y k(x,y) = \int i s e^{-i\langle s, x-y\rangle}\, \mathrm{d}\mu(s), \tag{9}$$

$$\nabla_x \cdot \nabla_y k(x,y) = \int \|s\|^2 e^{-i\langle s, x-y\rangle}\, \mathrm{d}\mu(s). \tag{10}$$

Now, let

$$\eta(x,s) = \frac{1}{p(x)}\left\{ e^{-i\langle s, x\rangle}\nabla q(x) - i s e^{-i\langle s, x\rangle} q(x)\right\} = \frac{q(x)}{p(x)}\left\{ e^{-i\langle s, x\rangle}\nabla \log q(x) - i s e^{-i\langle s, x\rangle}\right\}$$

and note through direct calculation and Equations (8) to (10) that

$$\int \eta(x,s)\cdot\overline{\eta(y,s)}\, \mathrm{d}\mu(s)$$

$$= \frac{q(x)}{p(x)}\frac{q(y)}{p(y)}\int \left\{ \begin{array}{c} \|s\|^2 + i s \cdot \nabla \log q(x) \\ -i s \cdot \nabla \log q(y) + \nabla \log q(x)\cdot \nabla \log q(y) \end{array}\right\} e^{-i\langle s, x-y\rangle}\, \mathrm{d}\mu(s)$$

$$= \frac{q(x)}{p(x)}\frac{q(y)}{p(y)}\left\{ \begin{array}{c} \nabla_x \cdot \nabla_y k(x,y) + \nabla_y k(x,y)\cdot \nabla \log q(x) \\ +\nabla_x k(x,y)\cdot \nabla \log q(y) + k(x,y)\nabla \log q(x)\cdot \nabla \log q(y) \end{array}\right\}$$

$$= \frac{q(x)}{p(x)}\frac{q(y)}{p(y)} k_q(x,y).$$

Thus, integrating with respect to $\pi$, and assuming that we may interchange the order of integrals, we have that

$$\int \left\| \int \eta(x,s)\, \mathrm{d}\pi(x)\right\|_{\mathbb{C}}^2 \mathrm{d}\mu(s) = \int \iint \eta(x,s)\cdot\overline{\eta(y,s)}\, \mathrm{d}\pi(x)\mathrm{d}\pi(y)\, \mathrm{d}\mu(s)$$

$$= \iint \int \eta(x,s)\cdot\overline{\eta(y,s)}\, \mathrm{d}\mu(s)\, \mathrm{d}\pi(x)\mathrm{d}\pi(y)$$

$$= \iint \frac{q(x)}{p(x)}\frac{q(y)}{p(y)} k_q(x,y)\, \mathrm{d}\pi(x)\mathrm{d}\pi(y) = \mathrm{D}_{p,q}(\pi)^2,$$

where the final equality is Proposition 5. This establishes the result. $\qquad\square$

To prove Theorem 1, two intermediate results are required:

**Proposition 7.** *For an element $\pi \in \mathcal{P}(\mathbb{R}^d)$, assume $Z := \int (q/p)\, \mathrm{d}\pi \in (0,\infty)$. Assume that $k$ and $\pi$ satisfy the preconditions of Proposition 2, and that $\int \|\nabla \log q\|^{\alpha/(\alpha-1)}(q/p)\, \mathrm{d}\pi < \infty$. Let $\bar{\pi} := (q\pi)/(pZ)$. Then $\bar{\pi} \in \mathcal{P}(\mathbb{R}^d)$ and*

$$\mathrm{D}_{p,q}(\pi) = Z\mathrm{D}_{q,q}(\bar{\pi}).$$

*Proof.* The assumption $Z \in (0,\infty)$ implies that $\bar{\pi} \in \mathcal{P}(\mathbb{R}^d)$. Furthermore, the assumption $\int \|\nabla \log q\|^{\alpha/(\alpha-1)}(q/p)\, \mathrm{d}\pi < \infty$ implies that $\int \|\nabla \log q\|^{\alpha/(\alpha-1)}\, \mathrm{d}\bar{\pi} < \infty$. Thus the assumptions of Proposition 2 are satisfied for both $\pi$ and $\bar{\pi}$, and thus both $\mathrm{D}_{p,q}(\pi)$ and $\mathrm{D}_{q,q}(\bar{\pi})$ are well-defined. Now, with $\xi$ as in Proposition 4, notice that

$$\mathrm{D}_{p,q}(\pi) = \|\xi\|_{\mathcal{H}(k)^s} = \left\| \int \frac{q(x)}{p(x)}\left[\nabla k(x,\cdot) + k(x,\cdot)\nabla \log q(x)\right] \mathrm{d}\pi(x)\right\|_{\mathcal{H}(k)^d}$$

$$= \left\| \int \left[\nabla k(x,\cdot) + k(x,\cdot)\nabla \log q(x)\right] Z\mathrm{d}\bar{\pi}(x)\right\|_{\mathcal{H}(k)^d} = Z\mathrm{D}_{q,q}(\bar{\pi}),$$

as claimed. $\qquad\square$

**Proposition 8.** *Let $f : \mathbb{R}^d \to [0,\infty)$ and $\pi \in \mathcal{P}(\mathbb{R}^d)$. Then $\int f^\alpha\, \mathrm{d}\pi > 0 \Rightarrow \int f\, \mathrm{d}\pi > 0$, for all $\alpha \in (0,\infty)$.*

*Proof.* From the definition of the Lebesgue integral, we have that $\int f^\alpha \, \mathrm{d}\pi = \sup\{\int s \, \mathrm{d}\pi : s$ a simple function with $0 \leq s \leq f^\alpha\} > 0$. Thus there exists a simple function $s = \sum_{i=1}^m s_i 1_{S_i}$ with $0 \leq s \leq f^\alpha$ and $\int s \, \mathrm{d}\pi > 0$. Here the $s_i \in \mathbb{R}$ and the measurable sets $S_i \subset \mathbb{R}^d$ are disjoint. In particular, it must be the case that at least one of the coefficients $s_i$ is positive; without loss of generality suppose $s_1 > 0$. Then $\tilde{s} := s_1^{1/\alpha} 1_{S_1}$ is a simple function with $0 \leq \tilde{s} \leq f$ and $\int \tilde{s} \, \mathrm{d}\pi > 0$. It follows that $\int f \, \mathrm{d}\pi = \sup\{\int s \, \mathrm{d}\pi : s$ a simple function with $0 \leq s \leq f\} > 0$. $\qquad\square$

*Proof of Theorem 1.* Since $\int (q/p)^\alpha \, \mathrm{d}\pi_n \in (0,\infty)$ and $q/p \geq 0$, from Proposition 8 we have, for each $n$, that $Z_n := \int q/p \, \mathrm{d}\pi_n > 0$. Thus the assumptions of Proposition 7 are satisfied by $k$ and each $\pi_n$, which guarantees that $\mathrm{D}_{p,q}(\pi_n) = Z_n \mathrm{D}_{q,q}(\bar{\pi}_n)$ where $\bar{\pi}_n := (q\pi_n)/(pZ_n) \in \mathcal{P}(\mathbb{R}^d)$.

Now, since $\mathrm{W}_1(\pi_n, p; q/p) \to 0$, taking $f = 1$ we obtain $Z_n = \int fq/p \, \mathrm{d}\pi_n \to \int fq/p \, \mathrm{d}p = 1$. In addition, note that

$$
\begin{aligned}
\mathrm{W}_1(\bar{\pi}_n, q) &= \sup_{L(f) \leq 1} \left| \int f \, \mathrm{d}\bar{\pi}_n - \int f \, \mathrm{d}q \right| \\
&= \sup_{L(f) \leq 1} \left| \int f(0) + [f(x) - f(0)] \, \mathrm{d}\bar{\pi}_n(x) - \int f(0) + [f(x) - f(0)] \, \mathrm{d}q(x) \right| \\
&= \sup_{L(f) \leq 1} \left| \int [f(x) - f(0)] \, \mathrm{d}\bar{\pi}_n(x) - \int [f(x) - f(0)] \, \mathrm{d}q(x) \right| \\
&= \sup_{\substack{L(f) \leq 1 \\ f(0)=0}} \left| \int f \, \mathrm{d}\bar{\pi}_n - \int f \, \mathrm{d}q \right|
\end{aligned}
$$

Thus, from the triangle inequality, we obtain the bound

$$
\begin{aligned}
\mathrm{W}_1(\bar{\pi}_n, q) &= \sup_{\substack{L(f) \leq 1 \\ f(0)=0}} \left| \int f \, \mathrm{d}\bar{\pi}_n - \int f \, \mathrm{d}q \right| \\
&= \sup_{\substack{L(f) \leq 1 \\ f(0)=0}} \left| \int \frac{fq}{pZ_n} \, \mathrm{d}\pi_n - \int \frac{fq}{p} \, \mathrm{d}p \right| \\
&\leq \sup_{\substack{L(f) \leq 1 \\ f(0)=0}} \left| \int \frac{fq}{pZ_n} \, \mathrm{d}\pi_n - \int \frac{fq}{p} \, \mathrm{d}\pi_n \right| + \sup_{\substack{L(f) \leq 1 \\ f(0)=0}} \left| \int \frac{fq}{p} \, \mathrm{d}\pi_n - \int \frac{fq}{p} \, \mathrm{d}p \right| \\
&= \underbrace{\left( \frac{1 - Z_n}{Z_n} \right)}_{\to 0} \underbrace{\sup_{\substack{L(f) \leq 1 \\ f(0)=0}} \left| \int \frac{fq}{p} \, \mathrm{d}\pi_n \right|}_{(*)} + \underbrace{\mathrm{W}_1\left( \pi_n, p; \frac{q}{p} \right)}_{\to 0}
\end{aligned}
$$

as $n \to \infty$. For $(*)$, since $q/p \geq 0$, the supremum is realised by $f(x) = \|x\|$ and

$$
(*) = \int \|x\| \frac{q(x)}{p(x)} \, \mathrm{d}\pi_n(x) < \infty.
$$

Thus we have established that $\mathrm{W}_1(\bar{\pi}_n, q) \to 0$. Since $\nabla \log q$ is Lipschitz with $\int \|\nabla \log q\|^2 \, \mathrm{d}q < \infty$ and $k$ has continuous and bounded second derivatives, the standard kernel Stein discrepancy has 1-Wasserstein convergence detection [Proposition 9 of Gorham and Mackey, 2017], meaning that $\mathrm{W}_1(\bar{\pi}_n, q) \to 0$ implies that $\mathrm{D}_{q,q}(\bar{\pi}_n) \to 0$ and thus, since $Z_n \to 1$, $\mathrm{D}_{p,q}(\pi_n) \to 0$. This completes the proof. $\qquad\square$

To prove Proposition 3, an intermediate result is required:

**Proposition 9.** *Let $k(x, y) = \phi(x - y)$ be a kernel with $\phi$ twice differentiable and let $q \in \mathcal{P}(\mathbb{R}^d)$ with $\nabla \log q$ well-defined. Then $k_q(x, x) = -\Delta\phi(0) + \phi(0)\|\nabla \log q(x)\|^2$, where $\Delta = \nabla \cdot \nabla$ and $k_q$ was defined in Proposition 5.*

*Proof.* First, note that we must have $\nabla\phi(0) = 0$, else the symmetry property of $k$ would be violated. Now, $\nabla_x k(x,y) = (\nabla\phi)(x-y)$, $\nabla_y k(x,y) = -(\nabla\phi)(x-y)$ and $\nabla_x \cdot \nabla_y k(x,y) = -\Delta\phi(x-y)$. Thus $\nabla_x k(x,y)|_{y=x} = \nabla_y k(x,y)|_{x=y} = 0$ and $\nabla_x \cdot \nabla_y k(x,y)|_{x=y} = -\Delta\phi(0)$. Plugging these expressions into Equation (5) yields the result. $\square$

*Proof of Proposition 3.* Let $k_q$ be defined as in Proposition 5. From Cauchy–Schwarz, we have that $k_q(x,y) \leq \sqrt{k_q(x,x)}\sqrt{k_q(y,y)}$, and plugging this into Proposition 5 we obtain the bound

$$D_{p,q}(\pi) \leq \int \frac{q(x)}{p(x)}\sqrt{k_q(x,x)}\,\mathrm{d}\pi(x) \tag{11}$$

For a radial kernel $k(x,y) = \phi(x-y)$ with $\phi$ twice differentiable, we have $\phi(0) > 0$ (else $k$ must be the zero kernel, since by Cauchy–Schwarz $|k(x,y)| \leq \sqrt{k(x,x)}\sqrt{k(y,y)} = \phi(0)$ for all $x,y \in \mathbb{R}^d$), and $k_q(x,x) = -\Delta\phi(0) + \phi(0)\|\nabla\log q(x)\|^2$ (from Proposition 9). Plugging this expression into Equation (11) and applying Jensen's inequality gives that

$$D_{p,q}(\pi)^2 \leq \int \frac{q(x)^2}{p(x)^2}\left[-\Delta\phi(0) + \phi(0)\|\nabla\log q(x)\|^2\right]\,\mathrm{d}\pi(x).$$

Now we may pick a choice of $p$, $q$ and $(\pi_n)_{n\in\mathbb{N}}$ ($\pi_n \overset{\mathrm{d}}{\nrightarrow} p$) for which this bound can be made arbitrarily small. One example is $q = \mathcal{N}(0,1)$, $p = \mathcal{N}(0,\sigma^2)$ (any fixed $\sigma > 1$), for which we have

$$D_{p,q}(\pi)^2 \leq \int \sigma^2 \exp(-\gamma\|x\|^2)\left[-\Delta\phi(0) + \phi(0)\|x\|^2\right]\,\mathrm{d}\pi(x)$$

where $\gamma = 1 - \sigma^{-2} > 0$. Then it is clear that, for example, the sequence $\pi_n = \delta(ne_1)$ (where $e_1 = [1,0,\ldots,0]^\top$) satisfies the assumptions of Proposition 2 and, for this choice,

$$D_{p,q}(\pi_n)^2 \leq \sigma^2 \exp(-\gamma n^2)\left[-\Delta\phi(0) + \phi(0)n^2\right] \to 0$$

and yet $\pi_n \overset{\mathrm{d}}{\nrightarrow} p$, as claimed. $\square$

*Proof of Theorem 2.* Since $\inf_{x\in\mathbb{R}^d} q(x)/p(x) > 0$, for each $n$ we have $Z_n := \int q/p\,\mathrm{d}\pi_n > 0$ and, furthermore, the assumption $\int \|\nabla\log q\|^{\alpha/(\alpha-1)}(q/p)\,\mathrm{d}\pi_n < \infty$ implies that $\int \|\nabla\log q\|^{\alpha/(\alpha-1)}\,\mathrm{d}\pi_n < \infty$. Thus the assumptions of Proposition 7 are satisfied by $k$ and each $\pi_n$, and thus we have $D_{p,q}(\pi_n) = Z_n D_{q,q}(\bar\pi_n)$ where $\bar\pi_n := (q\pi_n)/(pZ_n) \in \mathcal{P}(\mathbb{R}^d)$.

From assumption, $Z_n \geq \inf_{x\in\mathbb{R}^d} q(x)/p(x)$ is bounded away from 0. Thus if $D_{p,q}(\pi_n) \to 0$ then $D_{q,q}(\bar\pi_n) \to 0$. Furthermore, since $q \in \mathcal{Q}(\mathbb{R}^d)$ and the inverse multi-quadric kernel $k$ is used, the standard kernel Stein discrepancy has convergence control, meaning that $D_{q,q}(\bar\pi_n) \to 0$ implies $\bar\pi_n \overset{\mathrm{d}}{\to} q$ [Theorem 8 of Gorham and Mackey, 2017]. It therefore suffices to show that $\bar\pi_n \overset{\mathrm{d}}{\to} q$ implies $\pi_n \overset{\mathrm{d}}{\to} p$.

From the Portmanteau theorem, $\pi_n \overset{\mathrm{d}}{\to} p$ is equivalent to $\int g\,\mathrm{d}\pi_n \to \int g\,\mathrm{d}p$ for all functions $g$ which are continuous and bounded. Thus, for an arbitrary continuous and bounded function $g$, consider $f = gp/q$, which is also continuous and bounded. Then, since $\bar\pi_n \overset{\mathrm{d}}{\to} q$, we have (again from the Portmanteau theorem) that $Z_n^{-1}\int g\,\mathrm{d}\pi_n = \int f\,\mathrm{d}\bar\pi_n \to \int f\,\mathrm{d}q = \int g\,\mathrm{d}p$. Furthermore, the specific choice $g = 1$ shows that $Z_n^{-1} \to 1$, and thus $\int g\,\mathrm{d}\pi_n \to \int g\,\mathrm{d}p$ in general. Since $g$ was arbitrary, we have established that $\pi_n \overset{\mathrm{d}}{\to} p$, completing the proof. $\square$

To prove Theorem 3, an intermediate result is required:

**Proposition 10.** *Let $Q \in \mathcal{P}(\mathbb{R}^d)$ and let $k_q : \mathbb{R}^d \times \mathbb{R}^d \to \mathbb{R}$ be a reproducing kernel with $\int k_q(x,\cdot)\,\mathrm{d}q = 0$ for all $x \in \mathbb{R}^d$. Let $(x_n)_{n\in\mathbb{N}}$ be a sequence of random variables independently sampled from $q$ and assume that $\int \exp\{\gamma k_q(x,x)\}\,\mathrm{d}q(x) < \infty$ for some $\gamma > 0$. Then*

$$D_{q,q}\left(\frac{1}{n}\sum_{i=1}^{n}\delta(x_i)\right) \to 0$$

*almost surely as $n \to \infty$.*

*Proof.* This is Lemma 4 in Riabiz et al. [2022], specialised to the case where samples are independent and identically distributed. Although not identical to the statement in Riabiz et al. [2022], one obtains this result by following an identical argument and noting that the expectation of $k_q(x_i, x_j)$ is identically 0 when $i \neq j$ (due to independence of $x_i$ and $x_j$), so that bounds on these terms are not required. $\square$

*Proof of Theorem 3.* Since $\pi_n$ has finite support, all conditions of Theorem 2 are satisfied. Thus it is sufficient to show that almost surely $D_{p,q}(\pi_n) \to 0$. To this end, we follow Theorem 3 of Riabiz et al. [2022] and introduce the classical importance weights $w_i = p(x_i)/q(x_i)$, which are well-defined since $q > 0$. The normalised weights $\bar{w}_i = w_i/W_n$, $W_n := \sum_{j=1}^n w_j$ satisfy $0 \leq \bar{w}_1, \dots, \bar{w}_n$ and $\bar{w}_1 + \cdots + \bar{w}_n = 1$, and thus the optimality of $w^*$, together with the integral form of the GF-KSD in Equation (4), gives that

$$
D_{p,q}\left(\sum_{i=1}^n w_i^* \delta(x_i)\right) \leq D_{p,q}\left(\sum_{i=1}^n \bar{w}_i \delta(x_i)\right) = \frac{1}{W_n}\sqrt{\sum_{i=1}^n \sum_{j=1}^n k_q(x_i, x_j)}
$$

$$
= \left(\frac{1}{n} W_n\right)^{-1} D_{q,q}\left(\frac{1}{n}\sum_{i=1}^n \delta(x_i)\right) \qquad (12)
$$

From the strong law of large numbers, almost surely $n^{-1}W_n \to \int \frac{p}{q}\,\mathrm{d}q = 1$. Thus it suffices to show that the final term in Equation (12) converges almost surely to 0. To achieve this, we can check the conditions of Proposition 10 are satisfied.

Since $q$ and $h(\cdot) = \mathcal{S}_{p,q}k(x, \cdot)$ satisfy the conditions of Proposition 1, the condition $\int k_q(x, \cdot)\,\mathrm{d}q = 0$ is satisfied. Let $\phi(z) = (1 + \|z\|^2)^{-\beta}$ so that $k(x, y) = \phi(x - y)$ is the inverse multi-quadric kernel. Note that $\phi(0) = 1$ and $\Delta\phi(0) = -d$. Then, from Proposition 9, we have that $k_q(x, x) = -\Delta\phi(0) + \phi(0)\|\nabla \log q\|^2 = d + \|\nabla \log q\|^2$. Then

$$
\int \exp\{\gamma k_q(x, x)\}\,\mathrm{d}q(x) = \exp\{\gamma d\}\int \exp\{\gamma\|\nabla \log q\|^2\}\,\mathrm{d}q < \infty,
$$

which establishes that the conditions of Proposition 10 are satisfied and completes the proof. $\square$

## B  Stein Variational Inference Without Second-Order Gradient

This section contains sufficient conditions for unbiased stochastic gradient estimators to exist in the context of Stein Variational Inference; see Section 4.2 of the main text. The main result that we prove is as follows:

**Proposition 11** (Stochastic Gradients). *Let $p, q, R \in \mathcal{P}(\mathbb{R}^d)$ and $T^\theta : \mathbb{R}^d \to \mathbb{R}^d$ for each $\theta \in \mathbb{R}^p$. Let $\theta \mapsto \nabla_\theta T^\theta(x)$ be bounded. Assume that for each $\vartheta \in \mathbb{R}^p$ there is an open neighbourhood $N_\vartheta \subset \mathbb{R}^p$ such that*

$$
\int \sup_{\theta \in N_\vartheta} \left(\frac{q(T^\theta(x))}{p(T^\theta(x))}\right)^2 \mathrm{d}R(x) < \infty,
$$

$$
\int \sup_{\theta \in N_\vartheta} \frac{q(T^\theta(x))}{p(T^\theta(x))}\|\nabla \log r(T^\theta(x))\|\,\mathrm{d}R(x) < \infty,
$$

$$
\int \sup_{\theta \in N_\vartheta} \frac{q(T^\theta(x))}{p(T^\theta(x))}\|\nabla^2 \log r(T^\theta(x))\|\,\mathrm{d}R(x) < \infty,
$$

*for each of $r \in \{p, q\}$. Let $k$ be the inverse multi-quadric kernel in Equation (2) and let $u(x, y)$ denote the integrand in Equation (6). Then*

$$
\nabla_\theta D_{p,q}(\pi_\theta)^2 = \mathbb{E}\left[\frac{1}{n(n-1)}\sum_{i \neq j} \nabla_\theta u(T^\theta(x_i), T^\theta(x_j))\right]
$$

*where the expectation is taken with respect to independent samples $x_1, \dots, x_n \sim R$.*

The role of Proposition 11 is to demonstrate how an unbiased gradient estimator may be constructed, whose computation requires first-order derivatives of $p$ only, and whose cost is $O(n^2)$. Although the assumption that $\theta \mapsto \nabla_\theta T^\theta(x)$ is bounded seems strong, it can typically be satisfied by re-parametrisation of $\theta \in \mathbb{R}^p$.

To prove Proposition 11, we exploit the following general result due to Fisher et al. [2021]:

**Proposition 12.** *Let $R \in \mathcal{P}(\mathbb{R}^p)$. Let $\Theta \subseteq \mathbb{R}^p$ be an open set and let $u : \mathbb{R}^d \times \mathbb{R}^d \to \mathbb{R}$, $T^\theta : \mathbb{R}^d \to \mathbb{R}^d$, $\theta \in \Theta$, be functions such that, for all $\vartheta \in \Theta$,*

(A1) $\iint |u(T^\vartheta(x), T^\vartheta(y))| \, \mathrm{d}R(x)\mathrm{d}R(y) < \infty$;

(A2) *there exists an open neighbourhood $N_\vartheta \subset \Theta$ of $\vartheta$ such that*

$$\iint \sup_{\theta \in N_\vartheta} \|\nabla_\theta u(T^\theta(x), T^\theta(y))\| \, \mathrm{d}R(x)\mathrm{d}R(y) < \infty.$$

*Then $F(\theta) := \iint u(T^\theta(x), T^\theta(y)) \, \mathrm{d}R(x)\mathrm{d}R(y)$ is well-defined for all $\theta \in \Theta$ and*

$$\nabla_\theta F(\theta) = \mathbb{E}\left[ \frac{1}{n(n-1)} \sum_{i \neq j} \nabla_\theta u(T^\theta(x_i), T^\theta(x_j)) \right],$$

*where the expectation is taken with respect to independent samples $x_1, \dots, x_n \sim R$.*

*Proof.* This is Proposition 1 in Fisher et al. [2021]. $\qquad\square$

*Proof of Proposition 11.* In what follows we aim to verify the conditions of Proposition 12 hold for the choice $u(x, y) = k_q(x, y)$, where $k_q$ was defined in Equation (5).

*(A1):* From the first line in the proof of Proposition 4, the functions $\mathcal{S}_{p,q}k(x, \cdot)$, $x \in \mathbb{R}^d$, are in $\mathcal{H}(k)^d$. Since $(x, y) \mapsto u(x, y) = \langle \mathcal{S}_{p,q}k(x, \cdot), \mathcal{S}_{p,q}k(y, \cdot) \rangle_{\mathcal{H}(k)^d}$ is positive semi-definite, from Cauchy–Schwarz, $|u(x, y)| \leq \sqrt{u(x, x)}\sqrt{u(y, y)}$. Thus

$$\int |u(T^\theta(x), T^\theta(y))| \, \mathrm{d}R(x)\mathrm{d}R(y) = \int |u(x, y)| \, \mathrm{d}T^\theta_\# R(x)\mathrm{d}T^\theta_\# R(y)$$
$$\leq \left( \int \sqrt{u(x, x)} \, \mathrm{d}T^\theta_\# R(x) \right)^2.$$

Since $k(x, y) = \phi(x - y)$, we have from Proposition 9 that

$$u(x, x) = \left( \frac{q(x)}{p(x)} \right)^2 \left[ -\Delta\phi(0) + \phi(0)\|\nabla \log q(x)\|^2 \right]$$

and

$$\int \sqrt{u(x, x)} \, \mathrm{d}T^\theta_\# R(x) \leq \sqrt{\int \left( \frac{q}{p} \right)^2 \mathrm{d}T^\theta_\# R} \sqrt{\int |-\Delta\phi(0) + \phi(0)\|\nabla \log q\|^2| \, \mathrm{d}T^\theta_\# R}$$

which is finite by assumption.

*(A2):* Fix $x, y \in \mathbb{R}^d$ and let $R_x(\theta) := q(T^\theta(x))/p(T^\theta(y))$. From repeated application of the product rule of differentiation, we have that

$$\nabla_\theta u(T^\theta(x), T^\theta(y)) = \underbrace{k_q(T^\theta(x), T^\theta(y))\nabla_\theta [R_x(\theta)R_y(\theta)]}_{(*)} + \underbrace{R_x(\theta)R_y(\theta)\nabla_\theta k_q(T^\theta(x), T^\theta(y))}_{(**)}.$$

Let $b_p(x) := \nabla \log p(x)$, $b_q(x) := \nabla \log q(x)$, $b(x) := b_q(x) - b_p(x)$, and $[\nabla_\theta T^\theta(x)]_{i,j} = (\partial/\partial\theta_i)T^\theta_j(x)$. In what follows, we employ a matrix norm on $\mathbb{R}^{d \times d}$ which is consistent with the Euclidean norm on $\mathbb{R}^d$, meaning that $\|\nabla_\theta T^\theta(x)b(T^\theta(x))\| \leq \|\nabla_\theta T^\theta(x)\|\|b(T^\theta(x))\|$ for each $\theta \in \Theta$ and $x \in \mathbb{R}^d$. Considering the first term $(*)$, further applications of the chain rule yield that

$$\nabla_\theta [R_x(\theta)R_y(\theta)] = R_x(\theta)R_y(\theta)[\nabla_\theta T^\theta(x)b(T^\theta(x)) + \nabla_\theta T^\theta(y)b(T^\theta(y))]$$

and from the triangle inequality we obtain a bound

$$\|\nabla_\theta \left[ R_x(\theta) R_y(\theta) \right]\| \leq R_x(\theta) R_y(\theta) \left[ \|\nabla_\theta T^\theta(x)\| \|b(T^\theta(x))\| + \|\nabla_\theta T^\theta(y)\| \|b(T^\theta(y))\| \right].$$

Let $\lesssim$ denote inequality up to an implicit multiplicative constant. Since we assumed that $\|\nabla_\theta T^\theta(x)\|$ is bounded, and the inverse multi-quadric kernel $k$ is bounded, we obtain that

$$|(*)| \lesssim R_x(\theta) R_y(\theta) \left[ \|b(T^\theta(x))\| + \|b(T^\theta(y))\| \right].$$

Similarly, from Equation (5), and using also the fact that the inverse multi-quadric kernel $k$ has derivatives or all orders [Lemma 4 of Fisher et al., 2021], we obtain a bound

$$\|\nabla_\theta k_q(T^\theta(x), T^\theta(y))\| \lesssim \left[ 1 + \|b_q(T^\theta(x))\| + \|\nabla b_q(T^\theta(x))\| \right] \left[ 1 + \|b_q(T^\theta(y))\| \right]$$
$$+ \left[ 1 + \|b_q(T^\theta(y))\| + \|\nabla b_q(T^\theta(y))\| \right] \left[ 1 + \|b_q(T^\theta(x))\| \right]$$

which we multiply by $R_x(\theta) R_y(\theta)$ to obtain a bound on $(**)$. Thus we have an overall bound

$$\|\nabla_\theta u(T^\theta(x), T^\theta(y))\| \lesssim R_x(\theta) R_y(\theta) \left\{ \left[ 1 + \|b_q(T^\theta(x))\| + \|\nabla b_q(T^\theta(x))\| \right] \left[ 1 + \|b_q(T^\theta(y))\| \right] \right.$$
$$\left. + \left[ 1 + \|b_q(T^\theta(y))\| + \|\nabla b_q(T^\theta(y))\| \right] \left[ 1 + \|b_q(T^\theta(x))\| \right] \right\}.$$

Substituting this bound into $\iint \sup_{\theta \in N_\vartheta} \|\nabla_\theta u(T^\theta(x), T^\theta(y))\| \, dR(x) dR(y)$, and factoring terms into products of single integrals, we obtain an explicit bound on this double integral in terms of the following quantities (where $r \in \{p, q\}$):

$$\int \sup_{\theta \in N_\vartheta} R_x(\theta) \, dR(x)$$

$$\int \sup_{\theta \in N_\vartheta} R_x(\theta) \|b_r(T^\theta(x))\| \, dR(x)$$

$$\int \sup_{\theta \in N_\vartheta} R_x(\theta) \|\nabla b_r(T^\theta(x))\| \, dR(x)$$

which we have assumed exist.

Thus the conditions of Proposition 12 hold, and the result immediately follows. □

# C  Experimental Details

These appendices contain the additional empirical results referred to in Section 3, together with full details required to reproduce the experiments described in Section 4 of the main text.

## C.1  Detection of Convergence and Non-Convergence

This appendix contains full details for the convergence plots of Figure 1. In Figure 1, we considered the target distribution

$$p(x) = \sum_{i=1}^{3} w_i \mathcal{N}(x; \mu_i, \sigma_i^2),$$

where $\mathcal{N}(x; \mu, \sigma^2)$ is the univariate Gaussian density with mean $\mu$ and variance $\sigma^2$. The parameter choices used were $(w_1, w_2, w_3) = (0.375, 0.5625, 0.0625)$, $(\mu_1, \mu_2, \mu_3) = (-0.4, 0.3, 0.06)$ and $(\sigma_1^2, \sigma_2^2, \sigma_3^2) = (0.2, 0.2, 0.9)$.

The approximating sequences considered were location-scale sequences of the form $L_\#^n u$, where $L^n(x) = a_n + b_n x$ for some $(a_n)_{n \in \mathbb{N}}$ and $(b_n)_{n \in \mathbb{N}}$ and $u \in \mathcal{P}(\mathbb{R})$. For the converging sequences, we set $u = p$ and for the non-converging sequences, we set $u = \mathcal{N}(0, 0.5)$. We considered three different choices of $(a_n)_{n \in \mathbb{N}}$ and $(b_n)_{n \in \mathbb{N}}$, one for each colour. The sequences $(a_n)_{n \in \mathbb{N}}$ and $(b_n)_{n \in \mathbb{N}}$ used are shown in Figure S1. The specification of our choices of $q$ is the following:

- `Prior`: We took $q \sim \mathcal{N}(0, 0.75^2)$.
- `Laplace`: The Laplace approximation computed was $q \sim \mathcal{N}(0.3, 0.2041^2)$.

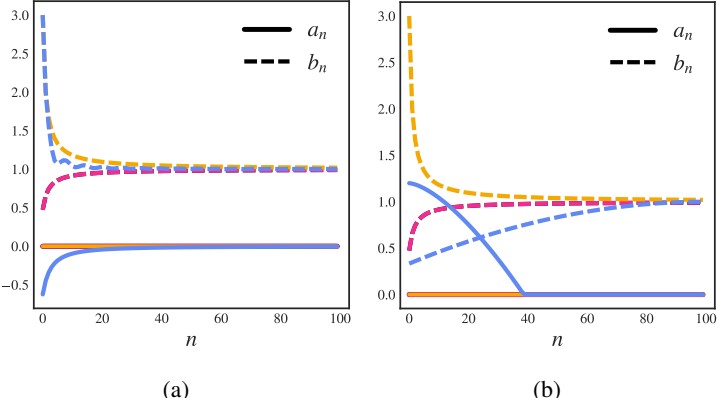

Figure S1: The sequences $a_n$ and $b_n$ used in the location-scale sequences. (a) The sequences $a_n$ and $b_n$ used in the location-scale sequences in Figure 1. (b) The sequences $a_n$ and $b_n$ used in the location-scale sequences in Figure 2. In each case, the colour used of each curve indicates which of the sequences $(\pi)_{n \in \mathbb{N}}$ they correspond to.

- GMM: The Gaussian mixture model was computed using $100$ samples from the target $p$. The number of components used was 2, since this value minimised the Bayes information criterion [Schwarz, 1978].

- KDE: The kernel density estimate was computed using $100$ samples from the target $p$. We utilised a Gaussian kernel $k(x, y) = \exp(-(x - y)^2/\ell^2)$ with the lengthscale or bandwidth parameter $\ell$ determined by Silverman's rule of thumb [Silverman, 1986].

The values of GF-KSD reported in Figure 1 were computed using a quasi Monte Carlo approximation to the integral (6), utilising a length 300 low-discrepancy sequence. The low discrepancy sequences were obtained by first specifying a uniform grid over $[0, 1]$ and then performing the inverse CDF transform for each member of the sequence $\pi_n$.

## C.2 Avoidance of Failure Modes

This appendix contains full details of the experiment reported in Section 3.3 and an explanation of the failure mode reported in Figure 2a. The sequences considered are displayed in Figure S2. Each sequence was a location-scale sequences of the form $L^n_{\#} u$, where $L^n(x) = a_n + b_n x$ for some $(a_n)_{n \in \mathbb{N}}$ and $(b_n)_{n \in \mathbb{N}}$ and $u \in \mathcal{P}(\mathbb{R})$. For the converging sequences, we set $u = p$. The specification of the settings of each failure mode are as follows:

- Failure mode (a) [Figure 2a]: We took $p$ as the target used in Figure 1 and detailed in Appendix C.1 and took $q \sim \mathcal{N}(0, 1.5^2)$. The $a_n$ and $b_n$ sequences used are displayed in Figure S1a. The values of GF-KSD reported were computed using a quasi Monte Carlo approximation, using a length 300 low discrepancy sequence. The low discrepancy sequences were obtained by first specifying a uniform grid over $[0, 1]$ and then performing the inverse CDF transform for each member of the sequence $\pi_n$.

- Failure mode (b) [Figure 2b]: We took $p \sim \mathcal{N}(0, 1)$ and $q \sim \mathcal{N}(-0.7, 0.1^2)$. The $a_n$ and $b_n$ sequences used are displayed in Figure S1b. The values of GF-KSD reported were computed using a quasi Monte Carlo approximation, using a length 300 low discrepancy sequence. The low discrepancy sequences were obtained by first specifying a uniform grid over $[0, 1]$ and then performing the inverse CDF transform for each member of the sequence $\pi_n$.

- Failure Mode (c) [Figure 2c]: In each dimension $d$ considered, we took $p \sim \mathcal{N}(0, I)$ and $q \sim \mathcal{N}(0, 1.1I)$. The $a_n$ and $b_n$ sequences[5] used are displayed in Figure S1b. The values of GF-KSD reported were computed using a quasi Monte Carlo approximation, using a length $1,024$ Sobol sequence in each dimension $d$.

---

[5]Note that for $d > 1$, we still considered location-scale sequences of the form $L^n(x) = a_n + b_n x$.

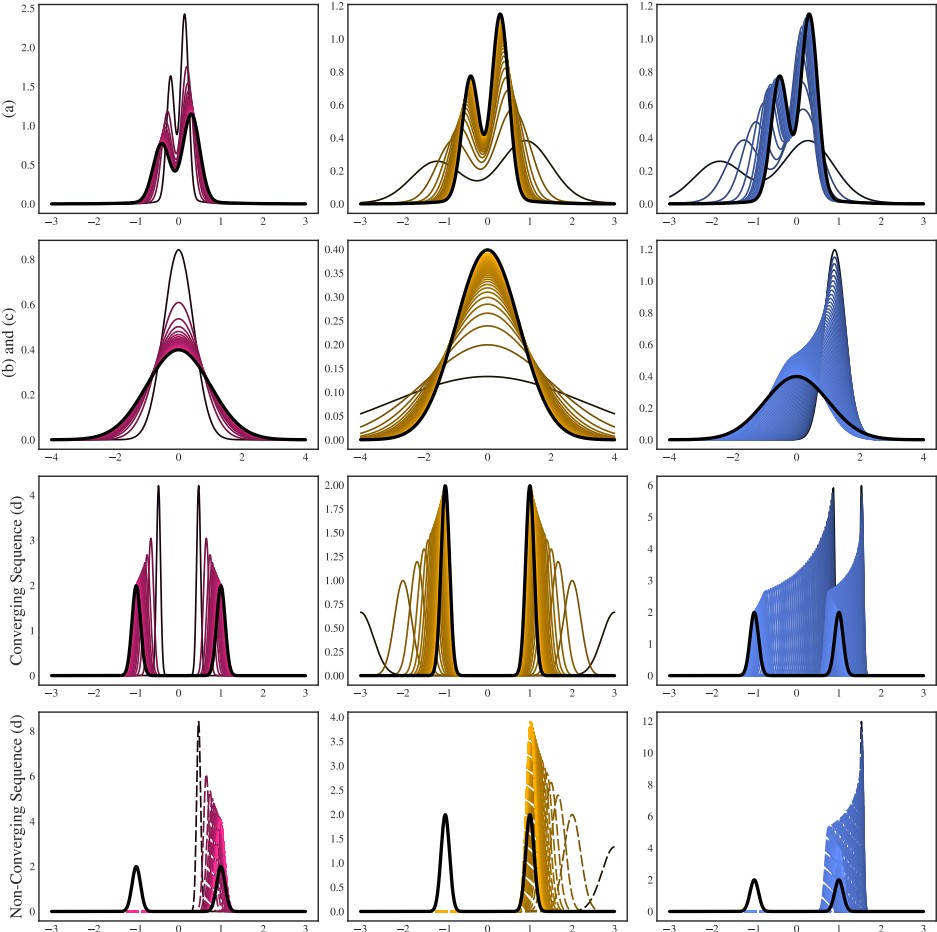

Figure S2: Test sequences $(\pi_n)_{n \in \mathbb{N}}$ used in Figure 2. The colour and style of each sequence indicates which of the curves in Figure 2 is being considered. In the second row from the top, the sequence used when $d = 1$ in Figure 2c is shown in the final column.

- Failure mode (d) [Figure 2d]: We took $p = q$, with $p(x) = 0.5 \mathcal{N}(x; -1, 0.1^2) + 0.5 \mathcal{N}(x; 1, 0.1^2)$, where $\mathcal{N}(x; \mu, \sigma^2)$ is the univariate Gaussian density with mean $\mu$ and variance $\sigma^2$. The $a_n$ and $b_n$ sequences used are displayed in Figure S1b. For the non-converging sequences we took $u = \mathcal{N}(1, 0.1^2)$ and used the $a_n$ and $b_n$ sequences specified in Figure S1b. The values of GF-KSD reported were computed using a quasi Monte Carlo approximation, using a length 300 low discrepancy sequence. The low discrepancy sequences were obtained by first specifying a uniform grid over $[0, 1]$ and then performing the inverse CDF transform for each member of the sequence $\pi_n$.

In Figure S3, we provide an account of the degradation of convergence detection between $q = \texttt{Prior}$ considered in Figure 1 and $q = \mathcal{N}(0, 1.5^2)$ of Failure mode (a). In Figure S3a, it can be seen that the value of the integrals $\int (q/p)^2 \, \mathrm{d}\pi_n$ are finite for each element of the pink sequence $\pi_n$. However, in Figure S3b, it can be seen that the values of the integrals $\int (q/p)^2 \, \mathrm{d}\pi_n$ are infinite for the last members of the sequence $\pi_n$, thus violating a condition of Theorem 1.

## C.3 Gradient-Free Stein Importance Sampling

This appendix contains full details for the experiment reported in Section 4.1. We considered the following Lotka–Volterra dynamical system:

$$\dot{u}(t) = \alpha' u(t) - \beta' u(t) v(t), \qquad \dot{v}(t) = -\gamma' v(t) + \delta' u(t) v(t), \qquad (u(0), v(0)) = (u_0', v_0').$$

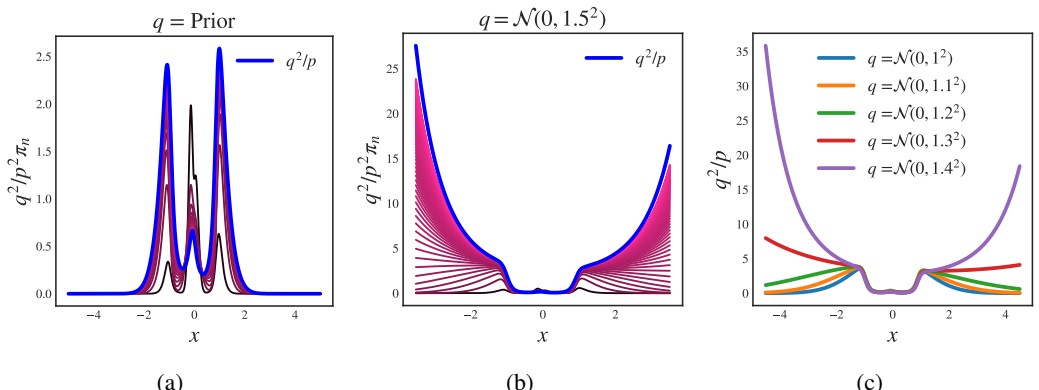

(a)                                    (b)                                    (c)

Figure S3: Explanation of Failure Mode (a). (a) Values of $(q/p)^2 \pi_n$ for $q = \texttt{Prior}$ and for the converging pink sequence displayed in Figure 1. (b) Values of $(q/p)^2 \pi_n$ for $q = \mathcal{N}(0, 1.5^2)$ and for the converging pink sequence displayed in the first column and first row of Figure S2. (c) Values of $q^2/p$ for different choices of $q$.

Using 21 observations $u_1, \ldots, u_{21}$ and $v_1, \ldots, v_{21}$ over times $t_1 < \ldots < t_{21}$, we considered the probability model

$$u_i \sim \text{Log-normal}(\log u(t_i), (\sigma_1')^2), \qquad v_i \sim \text{Log-normal}(\log v(t_i), (\sigma_2')^2).$$

In order to satisfy positivity constraints, we performed inference on the logarithm of the parameters $(\alpha, \beta, \gamma, \delta, u_0, v_0, \sigma_1, \sigma_2) = (\log \alpha', \log \beta', \log \gamma', \log \delta', \log u_0', \log v_0', \log \sigma_1', \log \sigma_2')$. We took the following independent priors on the constrained parameters:

$$\alpha' \sim \text{Log-normal}(\log(0.7), 0.6^2), \ \beta' \sim \text{Log-normal}(\log(0.02), 0.3^2),$$
$$\gamma' \sim \text{Log-normal}(\log(0.7), 0.6^2), \ \delta' \sim \text{Log-normal}(\log(0.02), 0.3^2),$$
$$u_0' \sim \text{Log-normal}(\log(10), 1), \ v_0' \sim \text{Log-normal}(\log(10), 1),$$
$$\sigma_1' \sim \text{Log-normal}(\log(0.25), 0.02^2), \ \sigma_2' \sim \text{Log-normal}(\log(0.25), 0.02^2).$$

In order to obtain independent samples from the posterior for comparison, we utilised Stan [Stan Development Team, 2022] to obtain $8,000$ posterior samples using four Markov chain Monte Carlo chains. Each chain was initialised at the prior mode. The data analysed are due to Hewitt [1921] and can be seen, along with a posterior predictive check, in Figure S4.

The Laplace approximation was obtained by the use of $48$ iterations of the L-BFGS optimisation algorithm [Liu and Nocedal, 1989] initialised at the prior mode. The Hessian approximation was obtained using Stan's default numeric differentiation of the gradient.

Finally, the quadratic programme defining the optimal weights of gradient-free Stein importance sampling (refer to Theorem 3) was solved using the splitting conic solver of O'Donoghue et al. [2016].

## C.4  Stein Variational Inference Without Second-Order Gradient

This appendix contains full details for the experiment reported in Section 4.2. We considered the following bivariate densities

$$p_1(x, y) := \mathcal{N}(x; 0, \eta_1^2) \, \mathcal{N}(y; \sin(ax), \eta_2^2),$$
$$p_2(x, y) := \mathcal{N}(x; 0, \sigma_1^2) \, \mathcal{N}(y; bx^2, \sigma_2^2),$$

where $\mathcal{N}(x; \mu, \sigma^2)$ is the univariate Gaussian density with mean $\mu$ and variance $\sigma^2$. The parameter choices for the sinusoidal experiment $p_1$ were $\eta_1^2 = 1.3^2, \eta_2^2 = 0.09^2$ and $a = 1.2$. The parameter choices for the banana experiment $p_2$ were $\sigma_1^2 = 1, \sigma_2^2 = 0.2^2$ and $b = 0.5$.

The development of a robust stochastic optimisation routine for measure transport with GF-KSD is beyond the scope of this work, and in what follows we simply report one strategy that was successfully used in the setting of the application reported in the main text. This strategy was based on

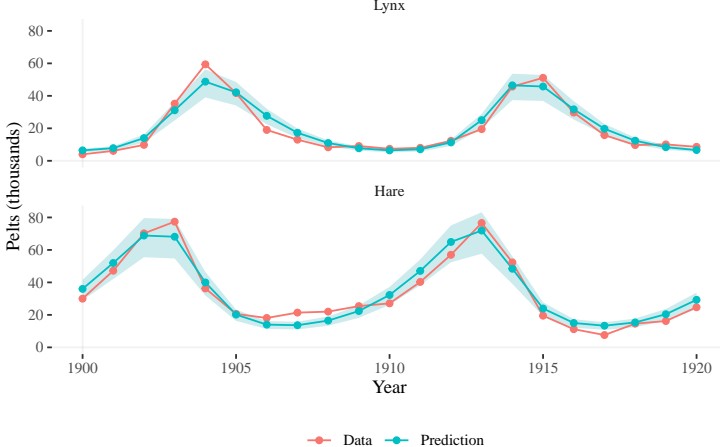

Figure S4: Posterior predictive check for the Lotka–Volterra model. The shaded blue region indicates the 50% interquartile range of the posterior samples.

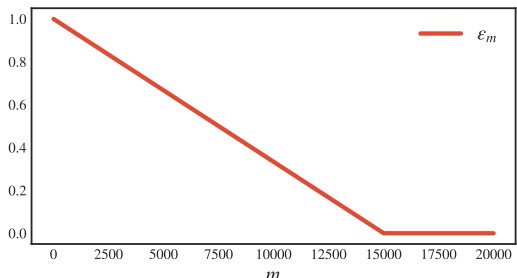

Figure S5: Tempering sequence $(\epsilon_m)_{m\in\mathbb{N}}$ used in each of the variational inference experiments.

tempering of $p$, the distributional target, to reduce a possibly rather challenging variational optimisation problem into a sequence of easier problems to be solved. Specifically, we considered tempered distributions $p_m \in \mathcal{P}(\mathbb{R}^d)$ with log density

$$\log p_m(x) = \epsilon_m \log p_0(x) + (1 - \epsilon_m) \log p(x),$$

where $(\epsilon_m)_{m\in\mathbb{N}} \in [0, 1]^{\mathbb{N}}$ is the tempering sequence and $p_0 \in \mathcal{P}(\mathbb{R}^d)$ is fixed. In this case $p_0$ was taken to be $\mathcal{N}(0, 2I)$ in both the banana and sinusoidal experiment. Then, at iteration $m$ of stochastic optimisation, we considered the variational objective function

$$\pi \mapsto \log D_{p_m, q}(\pi)$$

where $q = \pi_{\theta_m}$, as explained in the main text. Tempering has been applied in the context of normalising flows in Prangle and Viscardi [2023]. The tempering sequence used $(\epsilon_m)_{m\in\mathbb{N}}$ for each of the experiments is displayed in Figure S5.

For each experiment, the stochastic optimisation routine used was Adam [Kingma and Ba, 2015] with learning rate 0.001. Due to issues involving exploding gradients due to the $q/p$ term in GF-KSD, we utilised gradient clipping in each of the variational inference experiments, with the maximum 2-norm value taken to be 30. In both the banana and sinusoidal experiment, the parametric class of transport maps $T^\theta$ was the *inverse autoregressive flow* of Kingma et al. [2016]. In the banana experiment, the dimensionality of the hidden units in the underlying autoregressive neural network was taken as 20. In the sinusoidal experiment, the dimensionality of the hidden units in the underlying autoregressive neural network was taken as 30. For the comparison with standard kernel Stein discrepancy, the same parametric class $T^\theta$ and the same initialisations of $\theta$ were used.

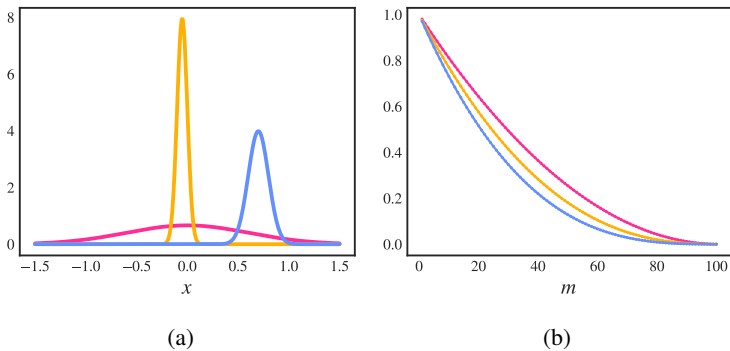

Figure S6: The $\pi_0$ and tempering sequences used in the additional convergence detection experiments. The colour of each curve indicates which of the sequences in Figure S7 and Figure S8 they correspond with. (a) The $\pi_0$ choices of each tempered sequence of distributions. (b) The tempering sequences $(\epsilon_m)_{m \in \mathbb{N}}$ considered.

## C.5 Additional Experiments

Appendix C.5.1 explores the impact of $p$ on the conclusions drawn in the main text. Appendix C.5.2 investigates the sensitivity of the proposed discrepancy to the choice of the parameters $\sigma$ and $\beta$ that appear in the kernel. Appendix C.5.3 compares the performance of GF-KSD importance sampling, KSD importance sampling and self-normalised importance sampling.

### C.5.1 Exploring the Effect of $p$

In this section we investigate the robustness of the convergence detection described in Figure 1 subject to different choices of the target $p$. We consider two further choices of $p$:

$$p_1(x) = \sum_{i=1}^{4} c_i \mathcal{N}(x; \mu_i, \sigma_i^2),$$

$$p_2(x) = \sum_{i=1}^{4} d_i \, \text{Student-T}(x; \nu, m_i, s_i),$$

where $\mathcal{N}(x; \mu, \sigma^2)$ is the univariate Gaussian density with mean $\mu$ and variance $\sigma^2$ and Student-T$(x; \nu, m, s)$ is the univariate Student-T density with degrees of freedom $\nu$, location parameter $m$ and scale parameter $s$. The parameter choices for $p_1$ were

$$(c_1, c_2, c_3, c_4) = (0.3125, 0.3125, 0.3125, 0.0625),$$
$$(\mu_1, \mu_2, \mu_3, \mu_4) = (-0.3, 0, 0.3, 0),$$
$$(\sigma_1^2, \sigma_2^2, \sigma_3^2, \sigma_4^2) = (0.1^2, 0.05^2, 0.1^2, 1).$$

The parameter choices for $p_2$ were $\nu = 10$ and

$$(d_1, d_2, d_3, d_4) = (0.1, 0.2, 0.3, 0.4),$$
$$(m_1, m_2, m_3, m_4) = (-0.4, -0.2, 0, 0.3),$$
$$(s_1, s_2, s_3, s_4) = (0.05, 0.1, 0.1, 0.3).$$

Instead of using the location-scale sequences of Figure 1, we instead considered tempered sequences of the form
$$\log \pi_n(x) = \epsilon_n \log \pi_0(x) + (1 - \epsilon_n) \log u(x).$$

For the converging sequences considered we set $u$ to be the target (either $u = p_1$ or $u = p_2$) and set $u = \mathcal{N}(x; 0, 0.4^2)$ for each of the non-converging sequences. The different sequences vary in choice of $\pi_0$ and tempering sequence $(\epsilon_n)_{n \in \mathbb{N}}$. These choices are displayed in Figure S6 and are taken as the same for both of the targets considered.

The specification of our choices of $q$ is the following:

- `Prior`: For $p_1$, we took $q \sim \mathcal{N}(0, 0.5^2)$. For $p_2$, we took $q \sim$ Student-T$(10, 0, 0.5)$.
- `Laplace`: For $p_1$, the Laplace approximation computed was $q \sim \mathcal{N}(0, 0.051^2)$. For $p_2$, the Laplace approximation computed was $q \sim \mathcal{N}(0, 0.125^2)$.
- `GMM`: For both targets, the Gaussian mixture model was computed using 100 samples from the target. In both cases, the number of components used was 3, since this value minimised the Bayes information criterion [Schwarz, 1978].
- `KDE`: For both targets, the kernel density estimate was computed using 100 samples from the target. In both cases, we utilised a Gaussian kernel $k(x, y) = \exp(-(x - y)^2/\ell^2)$ with the lengthscale or bandwidth parameter $\ell$ determined by Silverman's rule of thumb [Silverman, 1986].

Results for $p_1$ are displayed in Figure S7 and results for $p_2$ are displayed in Figure S8. It can be seen that for both target distributions and the different sequences considered, GF-KSD correctly detects convergence in each case. For both targets and for $q = $ `Laplace`, it can be seen that GF-KSD exhibits the same behaviour of Failure Mode (b), displayed in Figure 2b.

The values of GF-KSD reported in Figure S7 and Figure S8 were computed using a quasi Monte Carlo approximation to the integral (6), utilising a length 300 low-discrepancy sequence. Due to the lack of an easily computable inverse CDF, we performed an importance sampling estimate of GF-KSD as follows

$$\mathrm{D}_{p,q}(\pi) = \iint (\mathcal{S}_{p,q} \otimes \mathcal{S}_{p,q}) k(x, y) \, \mathrm{d}\pi(x) \, \mathrm{d}\pi(y)$$

$$= \iint (\mathcal{S}_{p,q} \otimes \mathcal{S}_{p,q}) k(x, y) \frac{\pi(x)\pi(y)}{w(x)w(y)} \, \mathrm{d}w(x) \, \mathrm{d}w(y),$$

where $w$ is the proposal distribution. For each element of a sequence $\pi_n$, we used a Gaussian proposal $w_n$ of the form:

$$\log w_n(x) = \epsilon_n \log \pi_0(x) + (1 - \epsilon_n) \log \mathcal{N}(x; 0, 0.4^2).$$

Since $\pi_0$ is Gaussian for each sequence, this construction ensures that each $w_n$ is both Gaussian and a good proposal distribution for $\pi_n$. The low-discrepancy sequences were then obtained by first specifying a uniform grid over $[0, 1]$ and the performing an inverse CDF transformation using $w_n$.

### C.5.2  Exploring the Effect of $\sigma$ and $\beta$

In this section we investigate the effect on convergence detection that results from changing the parameters $\sigma$ and $\beta$ in the inverse multi-quadric kernel (2). Utilising the same test sequences and choices of $q$ used in Figure 1, we plot the values of GF-KSD in Figure S9. It can be seen that the convergence detection is robust to changing values of $\sigma$ and $\beta$.

### C.5.3  GF-KSD vs. KSD Importance Sampling

In this section we investigate the performance of gradient-free Stein importance sampling, standard Stein importance sampling, and self-normalised importance sampling, as the distribution $q$ varies in quality as an approximation to $p$. We consider two different regimes:

1. $p = \mathcal{N}(0, I)$ and $q = \mathcal{N}(0, \lambda I)$ for $0.7 \leq \lambda \leq 1.3$.
2. $p = \mathcal{N}(0, I)$ and $q = \mathcal{N}(c\mathbf{1}, I)$ for $-0.6 \leq c \leq 0.6$, where $\mathbf{1} = (1, \ldots, 1)^\top$.

In both cases, we consider the performance of each approach for varying dimension $d$ and number of samples $n$. Results are reported in Figure S10 and Figure S11 for each regime respectively. The quadratic programme defining the optimal weights of gradient-free Stein importance sampling and Stein importance sampling (refer to Theorem 3) was solved using the splitting conic solver of O'Donoghue et al. [2016].

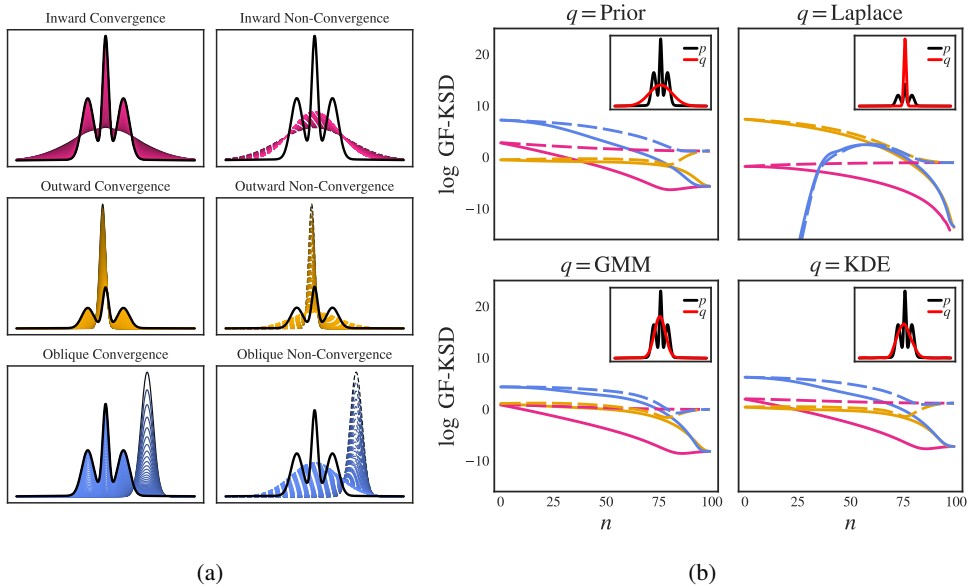

(a)            (b)

Figure S7: Additional empirical assessment of gradient-free kernel Stein discrepancy using the target $p_1$ defined in Appendix C.5.1. (a) Test sequences $(\pi_n)_{n\in\mathbb{N}}$, defined in Appendix C.5.1. The first column displays sequences (solid) that converge to the distributional target $p$ (black), while the second column displays sequences (dashed) which converge instead to a fixed Gaussian target. (b) Performance of gradient-free kernel Stein discrepancy, when different approaches to selecting $q$ are employed. The colour and style of each curve in (b) indicates which of the sequences in (a) is being considered. [Here we fixed the kernel parameters $\sigma = 1$ and $\beta = 1/2$.]

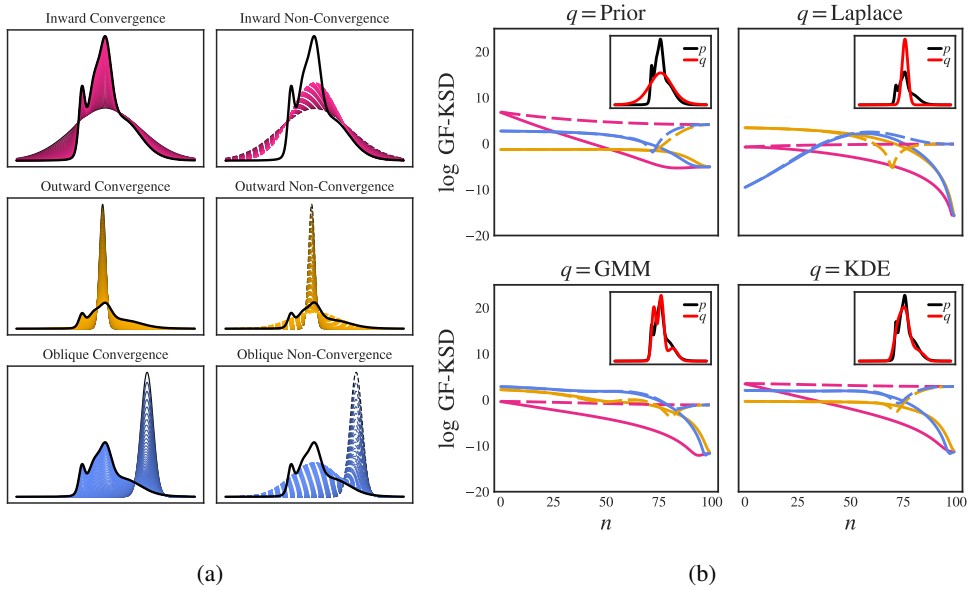

(a)            (b)

Figure S8: Additional empirical assessment of gradient-free kernel Stein discrepancy using the target $p_2$ defined in Appendix C.5.1. (a) Test sequences $(\pi_n)_{n\in\mathbb{N}}$, defined in Appendix C.5.1. The first column displays sequences (solid) that converge to the distributional target $p$ (black), while the second column displays sequences (dashed) which converge instead to a fixed Gaussian target. (b) Performance of gradient-free kernel Stein discrepancy, when different approaches to selecting $q$ are employed. The colour and style of each curve in (b) indicates which of the sequences in (a) is being considered. [Here we fixed the kernel parameters $\sigma = 1$ and $\beta = 1/2$.]

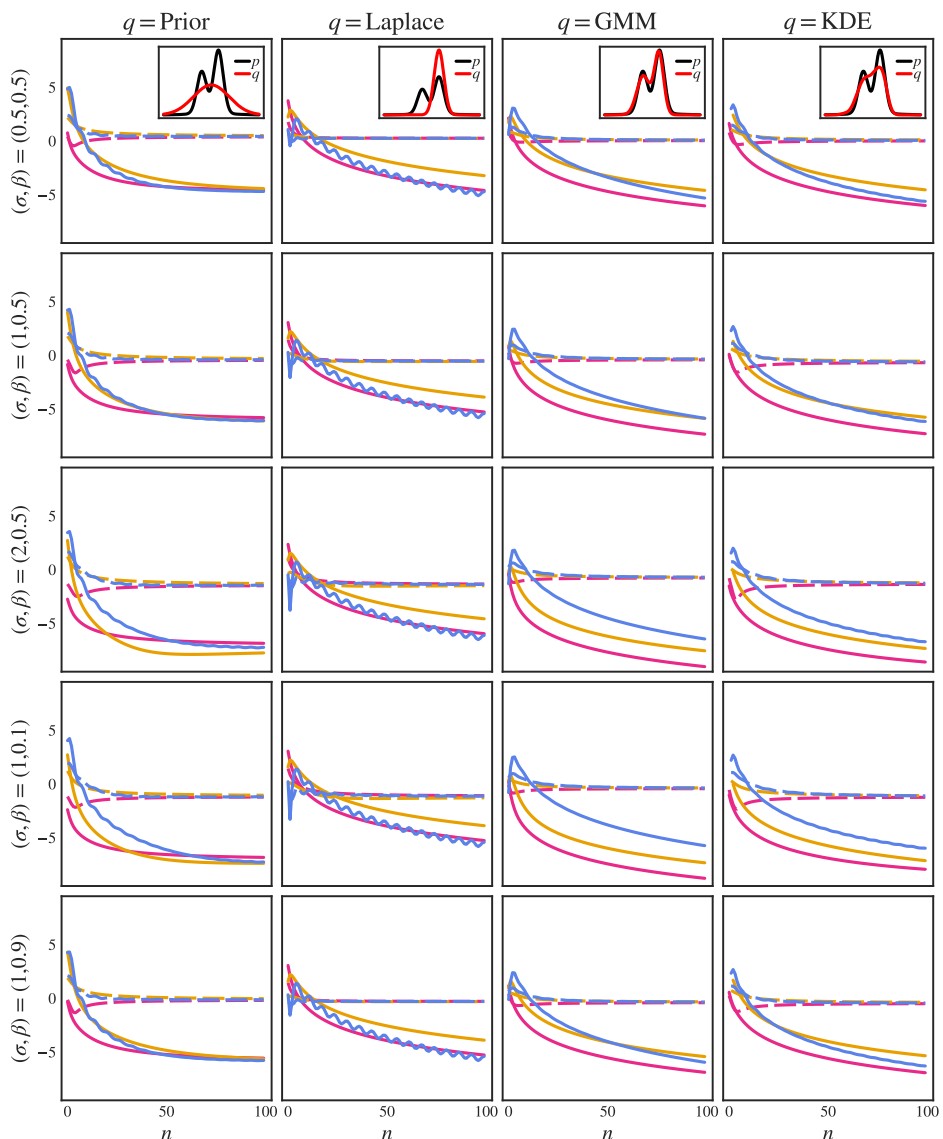

Figure S9: Comparison of different values of $\sigma$ and $\beta$ in the inverse multiquadric kernel. Here the vertical axis displays the logarithm of the gradient free kernel Stein discrepancy. The colour and style of each of the curves indicates which of the sequences in Figure 1 is being considered.

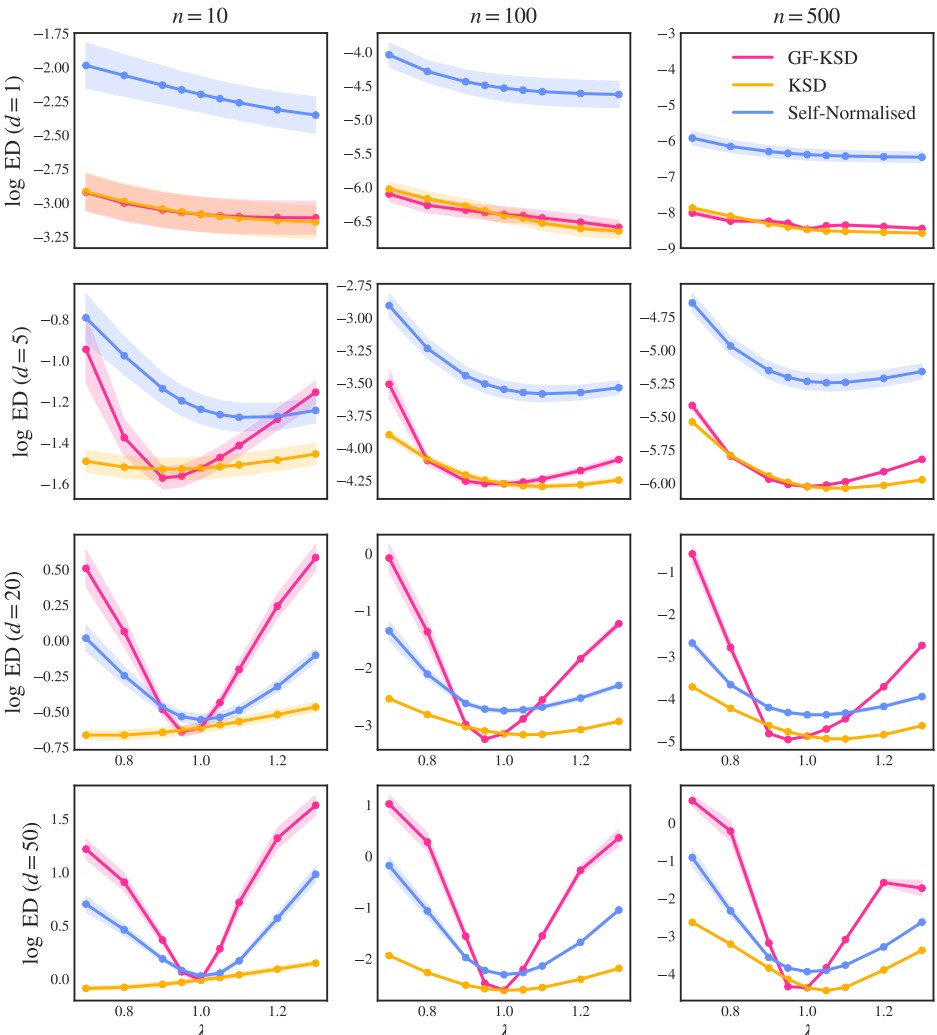

Figure S10: Comparison of the performance of importance sampling methodologies in varying dimension $d$ and number of sample points considered $n$ under the regime $q = \mathcal{N}(0, \lambda I)$. The approximation quality is quantified as the logarithm of the Energy Distance (ED).

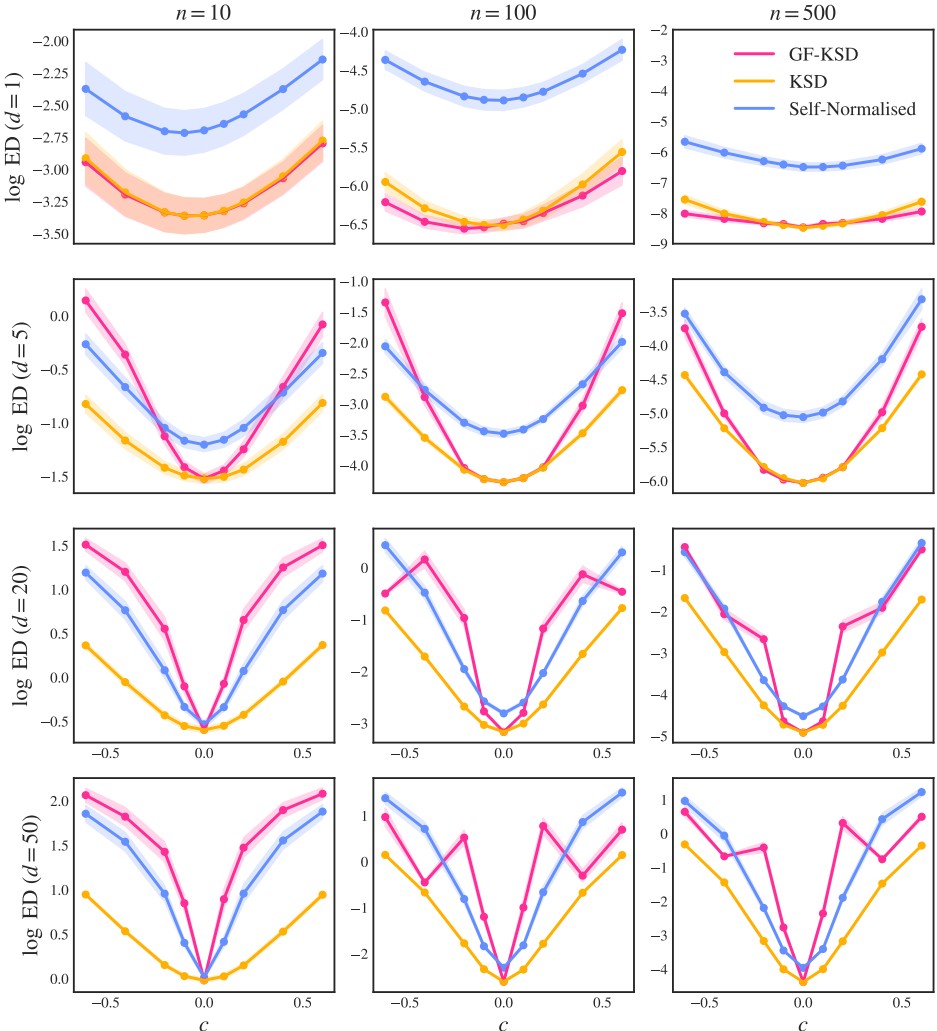

Figure S11: Comparison of the performance of importance sampling methodologies in varying dimension $d$ and number of sample points considered $n$ under the regime $q = \mathcal{N}(c, I)$. The approximation quality is quantified as the logarithm of the Energy Distance (ED).

