# OpenReview forum: "Gradient-Free Kernel Stein Discrepancy"
_NeurIPS.cc/2023/Conference — NeurIPS 2023 poster_

### Official Review · Reviewer_WN2C · 2023-06-19

**Soundness:** 2 fair
**Presentation:** 2 fair
**Contribution:** 2 fair
**Rating:** 4
**Confidence:** 3

**Summary:**

The paper provides a method to estimate the kernel Stein discrepancy without gradients.

**Strengths:**

The paper provides a method to estimate the kernel Stein discrepancy without gradients.

**Weaknesses:**

The paper is not clearly written for someone who is not familiar with the field. After reading, I'm still confused about whether someone else has published a similar method before.

**Questions:**

How limited is the class of distributions that allow gradient free estimation of the kernel Stein discrepancy?

And the paper probably cherry picked good experiments, how often do the mentioned failure modes occur?

**Limitations:**

The paper is not clearly written for someone who is not familiar with the field. After reading, I'm still confused about whether someone else has published a similar method before.

---

> ### Author Rebuttal · Authors · 2023-08-04
>
> Thank you for taking a look at our manuscript.
>
> Please see our global response to all Reviewers on the topic of how our manuscript has been presented.
>
> > After reading, I'm still confused about whether someone else has published a similar method before.
>
> Thank you for the opportunity to clarify this point.  A gradient-free Stein operator had been described in the context of Stein variational gradient descent (Han and Liu, 2018), but a Stein _discrepancy_ based on this operator had not been proposed or analysed.  Our work contributes the first such gradient-free Stein discrepancy, which is based on a reproducing kernel Hilbert space Stein set.  Reviewer DoGh has summarised the situation well, explaining that "whilst the gradient-free Stein operator has been previously discussed in the community (Han and Liu, 2018), its theoretical properties have not received sufficient attention. This paper established conditions under which the GF-KSD can detect and control convergence of a sequence of empirical distributions to a target probability measure (Theorem 1 and 2). To my knowledge, this contribution fills an important gap in the existing literature and significantly enhances our understanding of the topic".
>
> > How limited is the class of distributions that allow gradient free estimation of the kernel Stein discrepancy?
>
> For concreteness, we have interpreted this question as asking "for which $p$ does gradient-free kernel Stein discrepancy (KSD) offer convergence control?", since convergence control is the principal theoretical justification for using gradient-free KSD (GF-KSD) as an objective to be minimised.  Please let us know in the discussion if this was not what was being asked.
>
> If we choose $q = p$ we obtain the standard KSD, and in this case the class of $p$ for which KSD provided convergence control is known to include all distributions $p(x)$ on $\mathbb{R}^d$ with Lipschitz continuous $\nabla \log p(x)$ and _distantly dissipative_ tails (see Section 2.2).  The latter condition is implied when $p$ is strongly log-concave outside a compact set (i.e. sub-Gaussian tails).  For $q \neq p$, the distant dissipativity condition applies to $q$ rather than $p$, but then our $\inf_{x \in \mathbb{R}^d} q(x) / p(x) > 0$ condition in Theorem 2 implies that $p$ must also have sub-Gaussian tails in order for the preconditions of Theorem 2 to hold.  Thus, GF-KSD applies to distributions $p$ with sub-Gaussian tails -- this is more general than standard KSD since the requirement for $p$ to be distantly dissipative is removed.  We will make sure to emphasise this point in the revised manuscript.
>
> Thank you for your feedback; we hope that our clarifications, our commitment to improve the manuscript, and the remarks of Reviewer DoGh will have given you a more positive impression of our work.

---

### Official Review · Reviewer_DoGh · 2023-06-21

**Soundness:** 3 good
**Presentation:** 4 excellent
**Contribution:** 3 good
**Rating:** 8
**Confidence:** 3

**Summary:**

This paper explores the use of Stein discrepancies in scenarios where the score function of the target distribution is unavailable or computationally impractical to evaluate. The authors introduce a novel approach called gradient-free kernelized Stein discrepancy (GF-KSD), which leverages a Stein operator developed by Han and Liu (2018) that does not rely on the score function. The authors establish sufficient conditions for the resulting divergence to control and detect convergence. The empirical evaluation of this divergence extends its application to Stein importance sampling and Stein variational inference, surpassing the scope of the previous work by Han and Liu (2018).

**Strengths:**

**Theories**: Whilst the gradient-free Stein operator has been previously discussed in the community (Han and Liu, 2018), its theoretical properties have not received sufficient attention. This paper established conditions under which the GF-KSD can detect and control convergence of a sequence of empirical distributions to a target probability measure (Theorem 1 and 2). To my knowledge, this contribution fills an important gap in the existing literature and significantly enhances our understanding of the topic.

**Discussions**: Key results on both the theoretical and empirical sides are sufficiently discussed. Limitations of the GF-KSD are thoroughly examined and supported by empirical evidence (Section 3.3). Discussions on the choice of the degree of freedom, the density, are also included (Section 3.1).

**Structure**: This paper exhibits excellent writing with clear motivations throughout.

J. Han and Q. Liu. Stein variational gradient descent without gradient. In Proceedings of the 35th International Conference on Machine Learning, pages 1900–1908. PMLR, 2018.

**Weaknesses:**

**Experiments**: My only major concern is over the empirical results. The problems examined in the experiments are primarily toy examples with relatively low dimensionalities. where the one with the highest dimension is a 8-dimensional inference problem for a Lotka-Volterra model. The highest-dimensional problem considered is an 8-dimensional inference problem for a Lotka-Volterra model. To further validate the applicability of the proposed method in more realistic scenarios, additional empirical evidence on higher-dimensional problems would be beneficial.

Specifically, it would be interesting to investigate the performance of the default choice of Laplace approximation when the dimensionality of the target distribution is high. This analysis could shed light on whether the default approach continues to yield satisfactory results in non-toy scenarios.

Moreover, the reported numerical instability issue in Section 4.2 warrants further investigation, particularly in high-dimensional problems. Understanding the extent to which this instability becomes prominent in higher dimensions is crucial as it may impact the practical viability of the GF-KSD approach.

**Questions:**

It would be helpful to elaborate on the numerical instability issue noted in Section 4.2. Does this issue only occur when using GF-KSD for Stein variational inference, or does it also occur in the experiments in Section 4.1? Is that an artefact of the specific form of the target densities chosen for this experiment? Can this be avoided by a judicious choice of $q$?

**Limitations:**

Examples where the proposed approach can fail are extensively discussed in Section 3.3.

---

> ### Author Rebuttal · Authors · 2023-08-04
>
> Thank you for your kind comments and for your eloquent assessment of our manuscript.  As the reviewer most familiar with the literature on KSD, we sincerely hope that you will champion our work for acceptance, in light of the rather mixed scores we have received.
>
> > To further validate the applicability of the proposed method in more realistic scenarios, additional empirical evidence on higher-dimensional problems would be beneficial.
>
> We completely agree; as we mentioned to Reviewer 9g9f, our manuscript is limited to establishing the theoretical foundations of Gradient Free KSD (GF-KSD), enumerating its possible failure modes, and providing positive proofs of concept.  Subsequent follow-up work will be required to understand in detail the scenarios where GF-KSD can be effective and when it will encounter failure modes.  We will be sure to emphasise the need for a subsequent detailed empirical investigation in the conclusion section of our revised manuscript, including to problems that are higher-dimensional.
>
> > It would be helpful to elaborate on the numerical instability issue noted in Section 4.2. Does this issue only occur when using GF-KSD for Stein variational inference, or does it also occur in the experiments in Section 4.1? Is that an artefact of the specific form of the target densities chosen for this experiment? Can this be avoided by a judicious choice of $q$?
>
> Thank you for the opportunity to clarify this point:  The numerical issues that arose in Section 4.2 were encountered due to our simultaneous learning of both $p$, the distributional target, and $q$, the axuiliary distribution used to set up GF-KSD.  At early stages along the optimisation path, where the distribution $q$ is not yet an accurate reasonable of $p$, extreme values of the ratios $q/p$ were encountered and gradient clipping was required to regularise the stochastic gradient descent.  This problem was not present in Section 4.1, where $q$ was fixed as a Laplace approximation to $p$, such that extreme values of $q/p$ were not encountered.  These results are encouraging; GF-KSD is numerically stable in the context of Stein Importance Sampling when the Laplace approximation is used.  The use of GF-KSD in Stein Variational Inference requires further investigation; for example, there is at present no theoretical justification for the joint learning of $q$ along the optimisation path - we simply report that this numerical strategy was found to work well and represents "a promising avenue for further
> research".

---

> > ### Comment · Reviewer_DoGh · 2023-08-12
> >
> > Thank you for your response.
> >
> > With that said, I still think a non-toy numerical example would significantly improve the paper -- the proposed method is advertised to be a rescue when the score function of the statistical model is impractical to evaluate; despite the fact that GF-KSD is more practical than the standard KSD in these cases, for the presented examples many alternative methods exist and have been demonstrated to work reasonably well, making it unclear whether this method is practically useful (see also the comments by Reviewer 9g9f).
> >
> > Also, my concern of whether the default choice of Laplacian approximation can still give good performance in non-toy, high dimensional  examples is not yet answered.
> >
> > Due to the above, I have kept my scores, but happy to re-evaluate if the authors provided convincing numerical evidence to address these concerns before the rebuttal period ends.

---

### Official Review · Reviewer_9g9f · 2023-07-05

**Soundness:** 3 good
**Presentation:** 2 fair
**Contribution:** 2 fair
**Rating:** 4
**Confidence:** 4

**Summary:**

This paper proposed a posterior approximation using a new Stein discrepancy, which does not require derivatives of the statistical model. For that purpose, the authors derived the new discrepancy, called gradient-free KSD, and studied its statistical and convergence behaviors theoretically.
Then the authors developed algorithms for differential equations, for which stable computation of derivatives is difficult, and a new sampling algorithm that bypass the Hessian calculations.

**Strengths:**

- A new KSD that does not require gradients, similar to the idea of importance sampling, is proposed. This leads to new algorithms for differential equations where the gradient is difficult to compute and for problems requiring Hessian calculation.

- The authors studied the theoretical property of the proposed GF-KSD by extending the existing KSD theory.

- Not only theoretical analysis but also detailed numerical investigations of the choice of parameters and $q$ are carried out with the actual use in mind.



**Weaknesses:**

- The writing style is such that the main paper alone is not complete, and it is assumed that the reader will read the Appendix. For example, Eq. 6 of Line 115 does not appear in the main text, and the tilted Wasserstein distance defined in Theorem 1 is introduced without any explanation of its properties in the main text.

- The writing style could be improved since the discussion about existing research and the explanation of the proposed method are mixed, making the paper difficult to read.

-Some parts are mathematically undefined or under-discussed
 - In Definition 2, sup is undefined
 - In Line 158, at last, $\not \to$  is undefined.
 - I don't know how widely the tilted Wasserstein distance (TWD) in Theorem 1 is known to the general public, but there is no discussion of the properties of TWD. Therefore I could not understand how important Theorem 1 is, that is, how important it is when it is said that convergence of TWD leads to convergence of GF-KSD; even after reading the proof of Theorem 1, I could only understand that TWD is a convenient form of the usual Wasserstein distance, which is obtained after applying the triangle inequality.

- I could not understand the importance of the proposed method because I am not sure for what problems the proposed method is effective.
I agree that it may be useful for differential equation problems, but the authors only applied the method to very small models of Lotka-Volterra in the experiments. In such a setting, MCMC is the standard approach, and even if the model is high-dimensional, we can solve it efficiently by variational inference.
Also, although the combination with Stein variational inference seems interesting, I wondered if GF-KSD is really flexible enough to generate samples for complex real data under the two restrictions suggested in Section 3.
One restriction is that the tail of $q$ should not be far from the target distribution; the other is that it must not be high-dimensional.
I think the application to differential equations seems promising, so it would be better to find a problem setting where GF-KSD is more useful than MCMC and standard variational inference.

**Questions:**

I would appreciate it if the authors would answer the concerns described in Weakness.

As for minor questions;
- What do the dotted and solid lines in Figure 1(b) correspond to?

- Is it required to adjust parameters of the Laplace distribution, KDE, and GMM  for $q$ in some way ? If so, what is the recommended method ?

- Looking at Figure 3 (b), it seems that the number of samples ($n$) must be very large ($\log n=5$, i.e., $n=150$) even for low-dimensional problems such as d=8 in order for there to be any difference in energy distance. Is my understanding correct?

**Limitations:**

The limitation of the proposed method is discussed in detail.

---

> ### Author Rebuttal · Authors · 2023-08-04
>
> Thank you for taking a look at our manuscript.
>
> Please see our global response to all Reviewers on the topic of how our manuscript has been presented.
>
> > In Definition 2, sup is undefined
>
> The symbol "$\sup$" is the supremum; our understanding is that we can assume familiarity with "$\sup$" for NeurIPS, but we respectfully defer to the Area Chair for guidence on this point.
>
> > In Line 158, at last, $\nrightarrow$ is undefined.
>
> The symbol "$\nrightarrow$" is the logical opposite of $\rightarrow$, i.e. "does not converge"; our understanding is again that we can assume familiarity with "$\nrightarrow$" for NeurIPS, but we again respectfully defer to the Area Chair for guidence on this point.
>
> > I don't know how widely the tilted Wasserstein distance (TWD) in Theorem 1 is known to the general public, but there is no discussion of the properties of TWD. Therefore I could not understand how important Theorem 1 is, that is, how important it is when it is said that convergence of TWD leads to convergence of GF-KSD; even after reading the proof of Theorem 1, I could only understand that TWD is a convenient form of the usual Wasserstein distance, which is obtained after applying the triangle inequality.
>
> Thank you for the opportunity to clarify this point:  The TWD was introduced quite recently, in Proposition 3.3 of Huggins and Mackey 2018, so it is not widely known at present.  The special case where the tilting function $g(x) = 1$ is constant corresponds to the standard 1-Wasserstein distance.  Loosely speaking, for a tilting function $g(x)$ that is bounded away from $0$ on a compact subset $K \subset \mathbb{R}^d$, the topologies induced by TWD and 1-Wasserstein distances will be identical (i.e. convergence in one implies convergence in the other, and vice versa).  This is because the tilting function $g(x)$ can be viewed as importance weights, so that instead of comparing two measures $p(x)$ and $q(x)$ directly we compare instead their importance re-weighted measures, proportional to $g(x)p(x)$ and $g(x)q(x)$, and it is clear that convergence of either implies convergence of the other under compactness.  However, if $g(x)$ can approach either $0$ or $\infty$, which is what happens in our work concerning measures on $\mathbb{R}^d$, then the topologies of TWD and 1-Wasserstein distances do not coincide in general.
>
> On the other hand, we note that TWD induces a much weaker topology than, for example, divergences such as Kullback--Leibler or Hellinger, since it does not require absolute continuity of measures.  Further, we emphasise that Theorem 2 (convergence control) is really the main result of our manuscript, as opposed to Theorem 1 (convergence detection).  This is because convergence control justifies the design of algorithms that seek to minimise GF-KSD, guaranteeing the consistency of the approximations that are generated.
>
> Based on your useful feedback, we commit to expand our discussion of TWD in the manuscript to include the above points, which should broaden the accessibility of our manuscript.
>
> > I could not understand the importance of the proposed method because I am not sure for what problems the proposed method is effective [...] I wondered if GF-KSD is really flexible enough to generate samples for complex real data.
>
> This is an excellent question; our manuscript takes only a first step toward answering it, establishing the theoretical foundations of Gradient Free KSD (GF-KSD), enumerating its possible failure modes, and providing positive proofs of concept.  Subsequent follow-up work will be required to understand in detail the scenarios where GF-KSD can be effective and when it will encounter failure modes.  We will be sure to emphasise the need for a subsequent detailed empirical investigation in the conclusion section of our revised manuscript.
>
> > What do the dotted and solid lines in Figure 1(b) correspond to?
>
> In the caption of Figure 1 we state that "the colour and style of each curve in (b) indicates which of the sequences in (a) is being considered", but we will rephrase this to be explicit that the (e.g.) dashed blue curves in (b) correspond to the sequence of approximating distributions displayed as dashed blue curves in (a).
>
> > Is it required to adjust parameters of the Laplace distribution, KDE, and GMM for $q$ in some way? If so, what is the recommended method?
>
> Please allow us to emphasise that KDE and GMM are not part of any method that we propose, we simply used these to generate some examples of distributions $q$, for the purpose of illustration in Figure 1.  We will add additional emphasis on this point in the manuscript.  The Laplace approximation has no degrees of freedom to be specified.
>
> > Looking at Figure 3 (b), it seems that the number of samples ($n$) must be very large ($\log n = 5$, i.e., $n = 150$) even for low-dimensional problems such as $d=8$ in order for there to be any difference in energy distance. Is my understanding correct?
>
> That is completely correct, but we would argue that $n = 150$ is not a "very large" number of samples if we aim to accurately represent an 8-dimensional target.  Of course, "very large" will be application-specific, but if we were to take a Cartesian product of a grid of size just 3 in each dimension then we would need $n = 3^8 = 6561$ samples, which is much greater than $n = 150$.
>
> Thank you once again for your constructive suggestions on how our manuscript can be improved; we hope that our commitment to resolve them will be reflected in your updated reviewer scores.

---

### Official Review · Reviewer_SmKF · 2023-08-07

**Soundness:** 3 good
**Presentation:** 4 excellent
**Contribution:** 3 good
**Rating:** 6
**Confidence:** 3

**Summary:**

The authors study the kernel Stein discrepancy based on a Stein operator, introduced previously in [Liu 2018], which does not require access to the gradient of the target density. As their main theoretical result, they prove that, under certain assumptions on the target distribution (and the auxiliary distribution q) and the approximating sequence, the discrepancy controls weak convergence. The authors provide recommendations for the choice of the auxiliary distribution q in the construction of the discrepancy and present an experimental study which identifies certain failure modes of the discrepancy. They also discuss two applications to posterior approximation.

**Strengths:**

The idea of the authors to use the gradient-free Stein operator in order to define a new kind of KSD is interesting and worth studying. There are certainly a lot of situations where the evaluation of the gradient of the model is costly in practice. Giving the practitioners an option to avoid evaluating it when computing the KSD is certainly useful. The discussion of the properties of the new GF-KSD is very thorough and based on both theoretical results and a substantial experimental study. I particularly liked the the experiments revealing failure modes of GF-KSD as those are not necessarily captured by the theoretical results.

**Weaknesses:**

The main theoretical contribution of the paper seems to be provided by Theorem 2. However, I am a bit concerned about its real applicability. The authors require that the auxiliary distribution $q$ have tails heavier than the target $p$. On the other hand, they require that $q$ is distantly dissipative, which precludes the use of heavy-tailed $q$. This all means that $p$ must not be heavy-tailed if the assumptions of Theorem 2 are to be satisfied.

Moreover, the authors suggest four potential choices of $q$ (Prior, Laplace, GMM and KDE) but note that GMM and KDE are impractical in general. At the same time, the Prior method does not satisfy the assumptions of Theorem 2 for heavy-tailed priors. The Laplace method, on the other hand, seems to satisfy the assumptions of Theorem 2 only for targets which are sub-Gaussian.

**Questions:**

Related to what I wrote above, could the authors state clearly what class of targets $p$ their Theorem 2 applies to? Could they also try to characterise the class of targets $p$ for which one can easily construct a useful auxiliary distribution $q$ in practice (using one of the methods of section 3.1), such that $q$ and $p$ satisfy the assumptions of Theorem 2?

**Limitations:**

I believe the real applicabilty of the theoretical results presented by the authors is somewhat limited, to an extent that is not clearly acknowledged in the paper. But I am looking forward to reading the authors' response as perhaps I'm missing something.

---

> ### Author Rebuttal · Authors · 2023-08-08
>
> Thank you for your thoughtful feedback, especially at short notice, which is greatly appreciated.
>
> > This all means that $p$ must not be heavy-tailed if the assumptions of Theorem 2 are to be satisfied.
>
> This is completely correct - but we emphasise that the standard kernel Stein discrepancy (KSD) also has this requirement.  That is, the standard KSD is only guaranteed to have convergence control in settings where the target distribution $p$ is distantly dissipative (Theorem 8 of Gorham and Mackey, 2017), meaning that $p$ cannot be heavy tailed.  Despite this limitation, standard KSD has been widely used for applications such as sampling and variational inference, for which convergence control is required.  This is encouraging, in the sense that there are a wide variety of important problems for which $p$ is not heavy tailed and KSD has been successfully used.  There is active research on Stein's method for heavy tailed targets in the Probability community [for example, UB2022], but (as far as we are aware) the application of these techniques to the problem of posterior approximation has yet to be attempted.
>
> Thank you for raising this point - we will explicitly highlight the non-applicability of both KSD and GF-KSD to heavy-tailed $P$ in the revised manuscript, when discussing the preconditions of Theorem 2.
>
> > Related to what I wrote above, could the authors state clearly what class of targets $p$ their Theorem 2 applies to?  Could they also try to characterise the class of targets $p$ for which one can easily construct a useful auxiliary distribution $q$ in practice (using one of the methods of section 3.1), such that $q$ and $p$ satisfy the assumptions of Theorem 2?
>
> Thank you for the opportunity to clarify this point -- we first provide a mathematical answer, and then also a practical answer which addresses the fact that an appropriate and actionable choice of $q$ is required.
>
> Mathematical answer:
>
> If we choose $q=p$ we obtain the standard KSD, and in this case the class of $p$  for which KSD provided convergence control is known to include all distributions $p(x)$  on $\mathbb{R}^d$  with Lipschitz continuous $\nabla \log p(x)$  and _distantly dissipative_ tails (see Section 2.2). The latter condition is implied when $p$  is strongly log-concave outside a compact set (i.e. sub-Gaussian tails).
>
> For any $p$ (not necessarily distantly dissipative) with sub-Gaussian tails there exists a dominating Gaussian distribution $q$, i.e. with $\inf_{x \in \mathbb{R}^d} q(x) / p(x) > 0$, and since Gaussians are distantly dissipative Theorem 2 can be applied.  Thus GF-KSD is in principle _more widely applicable_ than standard KSD.
>
> We will make sure to emphasise this point in the revised manuscript.
>
> Practical answer:
>
> Of course, if $p$ is implicitly defined then it may not be clear how $q$ should be selected.  If $p$ itself is distantly dissipative then $q$ could be taken to be a tempered version of $p$ (i.e. $q(x) \propto p(x)^t$ for some $0 < t < 1$), which would automatically guarantee that $\inf_{x \in \mathbb{R}^d} q(x) / p(x) > 0$.  But this "solution" raises the question of how to take the gradient of $q$, if taking the gradient of $p$ is presumed to be difficult.  As a more practical strategy, we propose to take $q$ to be a Laplace approximation of $p$, in the hope that the curvature of $p$ at the mode is an accurate proxy for the tail decay of $p$.  This strategy was seen to perform well in the experiments reported in Sections 3.1 and 4.1, but one can of course construct examples where this strategy will fail.
>
> In preparatory work, we also considered taking $q$ to be a Laplace approximation with an "inflated" covariance matrix; a conservative choice aimed at guaranteeing the $\inf_{x \in \mathbb{R}^d} q(x) / p(x) > 0$ condition is satisfied.  This did not lead to improved performance in any of our experiments, so was omitted from the manuscript.
>
> Please allow us to emphasise that this is an instance of the fundamental problem that if $p$ is implicitly defined via Bayes' theorem, such that the mathematical properties (tail decay, curvature, etc) of $p$ are unknown to the user, then we cannot hope to predict how well _any_ sampling algorithm will work.  In specific applications where the prior and the likelihood are mathematically tractable, we may be able to _derive_ appropriate choices of $q$ using similar methodology that is used for constructing dominating measures for rejection sampling; this would be an interesting direction for further work.
>
> All of the above discussion will be incorporated into the revised manuscript.
>
> Thank you once again for your comments and questions, which we hope we have adequately addressed.  To summarise; the theoretical applicability of GF-KSD is strictly greater than standard KSD in terms of convergence control, but neither KSD nor GF-KSD are appropriate when $p$ is heavy-tailed.  The practical choice of $q$ requires some care, but a promising strategy based on Laplace approximation was found to work well in the experiments we performed.  We hope that our commitment to clarify these points in the manuscript will allow you to increase your score for our work.
>
> References:
>
> [UB2022] Upadhye NS and Barman K, 2022. A unified approach to Stein’s method for stable distributions. Probability Surveys, 19, pp.533-589.

---

> > ### Comment · Reviewer_SmKF · 2023-08-16
> > **Reply to the authors**
> >
> > Thank you very much for your answers. I have increased my rating of the paper.

---

### Author Rebuttal · Authors · 2023-08-04

__On the Presentation of our Manuscript__

Our manuscript is theoretical in nature and is written for readers who have some familiarity with the concept of kernel Stein discrepancy (KSD).  This is a small but important subsection of the NeurIPS community, who we believe will be the readers most interested in this work (e.g. KSD is currently being used in the community for applications such as goodness-of-fit testing, sampling, gradient estimation, and variational inference).  Excellent introductions to KSD already exist, such as "A Short Introduction to Kernelized Stein Discrepancy" on Qiang Liu's website, or "Measuring Sample Quality with Kernels" by Gorham and Mackey (ICML 2017); there is also now a Wikipedia page on "Stein Discrepancy".  For readers with this background, we believe our mansucript strikes an appropriate balance between precision and concision.  For example, Reviewer DoGh comments that "this paper exhibits excellent writing with clear motivations throughout".

---

### Decision · Program_Chairs · 2023-09-21

**Decision:**

Accept (poster)

**Comment:**

This is paper makes nice contribution to the KSD literature by developing the properties of a KSD that can be applied when it is impractical to compute model log density gradients. However, in addition to the points brought up by the reviewers, the authors should address a two other points in their camera-ready version:

1. Clarify that they did not "introduce" this gradient-free KSD. Rather, it was first proposed in Han and Liu [2018].
2. Clarify that the gradient-free KSD is essentially an importance-sampled KSD using p as the importance distribution.